# Oncogenic context shapes the fitness landscape of tumor suppression

Lily M. Blair [1,17], Joseph M. Juan[1,17], Lafia Sebastian[1], Vy B. Tran[1], Wensheng Nie[1], Gregory D. Wall[1], Mehmet Gerceker[1], Ian K. Lai[1], Edwin A. Apilado[1], Gabriel Grenot[1], David Amar[1,2,3], Giorgia Foggetti[4], Mariana Do Carmo[5], Zeynep Ugur[4], Debbie Deng[6], Alex Chenchik[6], Maria Paz Zafra[7,8,9], Lukas E. Dow [7,10,11], Katerina Politi[4,5,12], Jonathan J. MacQuitty[1], Dmitri A. Petrov[13,14], Monte M. Winslow[15,16], Michael J. Rosen [1] ✉ & Ian P. Winters [1] ✉

Tumors acquire alterations in oncogenes and tumor suppressor genes in an adaptive walk through the fitness landscape of tumorigenesis. However, the interactions between oncogenes and tumor suppressor genes that shape this landscape remain poorly resolved and cannot be revealed by human cancer genomics alone. Here, we use a multiplexed, autochthonous mouse platform to model and quantify the initiation and growth of more than one hundred genotypes of lung tumors across four oncogenic contexts: KRAS G12D, KRAS G12C, BRAF V600E, and EGFR L858R. We show that the fitness landscape is rugged—the effect of tumor suppressor inactivation often switches between beneficial and deleterious depending on the oncogenic context—and shows no evidence of diminishing-returns epistasis within variants of the same oncogene. These findings argue against a simple linear signaling relationship amongst these three oncogenes and imply a critical role for off-axis signaling in determining the fitness effects of inactivating tumor suppressors.

Adaptation by natural selection is the central mechanism of evolution and is at the core of some of the greatest challenges facing humanity: from loss of biodiversity to the spread of infectious disease, to cancer development and resistance to therapy[1–3]. Our capacity to overcome these challenges is dependent on our ability to predict the evolutionary paths taken by such complex and evolving systems. The "fitness landscape", a map between the genotype and fitness of a

biological entity, is a key concept in evolutionary genetics[4] that provides a framework to understand what kinds of evolutionary paths are possible in a given system.

Theoretical investigations have suggested that fitness landscapes can be broadly categorized as smooth or rugged. In smooth, "Mount Fuji"[5] landscapes, a mutation that is adaptive in one context will be adaptive in all other contexts. Thus, adaptation continues until a

[1]D2G Oncology, Mountain View, CA, USA. [2]Department of Biomedical Data Science, Stanford University School of Medicine, Stanford, CA, USA. [3]Department of Cardiovascular Medicine and the Cardiovascular Institute, Stanford University School of Medicine, Stanford, CA, USA. [4]Yale Cancer Center, Yale School of Medicine, New Haven, CT, USA. [5]Department of Pathology, Yale School of Medicine, New Haven, CT, USA. [6]Cellecta, Mountain View, CA, USA. [7]Sandra and Edward Meyer Cancer Center, Department of Medicine, Weill Cornell Medicine, New York, NY, USA. [8]Excellence Research Unit "Modeling Nature" (MNat), University of Granada, E-18016 Granada, Spain. [9]Instituto de Investigación Biosanitaria de Granada (ibs.GRANADA), E-18071 Granada, Spain. [10]Weill Cornell Graduate School of Medical Sciences, Weill Cornell Medicine, New York, NY, USA. [11]Department of Medicine, Weill Cornell Medicine, New York, NY, USA. [12]Section of Medical Oncology, Department of Internal Medicine, Yale School of Medicine, New Haven, CT, USA. [13]Department of Biology, Stanford University, Stanford, CA, USA. [14]Chan Zuckerberg BioHub, San Francisco, CA, USA. [15]Department of Genetics, Stanford University School of Medicine, Stanford, CA, USA. [16]Department of Pathology, Stanford University School of Medicine, Stanford, CA, USA. [17]These authors contributed equally: Lily M. Blair, Joseph M. Juan. ✉e-mail: mike@d2g-oncology.com; ian@d2g-oncology.com

unique fitness peak is reached. In contrast, rugged landscapes, characterized by epistatic interactions wherein the effect of one mutation depends upon others, contain multiple peaks with intervening valleys, inhibiting certain paths. Fitness landscapes can also vary in steepness; individual mutations can be strongly adaptive in steep landscapes while yielding smaller fitness gains in flatter landscapes.

Empirical studies of fitness landscapes have revealed two general observations. First, rugged fitness landscapes containing both pairwise and higher-order epistasis are common, making some evolutionary trajectories more viable than others[6]. Second, fixed adaptive mutations reduce the selective advantage of all subsequent mutations—a property termed diminishing-returns epistasis. Diminishing-returns epistasis was discovered in experimental evolution systems[7–9], but it remains unknown whether this phenomenon is generalizable across biological systems.

Cancer progression is a quintessential example of a walk on an adaptive fitness landscape, with tumor growth depending on the cooperation of multiple driver mutations[10–12]. While cancer genome sequencing has revealed a vast set of putative cancer drivers—oncogenes and tumor suppressors—mapping the tumor fitness landscape that emerges from coincident alteration of these genes remains a key gap in our understanding of cancer biology. While oncogene pairs are known not to have "Mount Fuji"-like additive fitness effects—indeed, the presence of more than one oncogene may even lead to growth arrest or apoptosis[13]—far less is known about the epistasis of oncogene-tumor suppressor pairs. Inferring this dimension of the fitness landscape is the focus of this manuscript.

A common approach to inferring epistatic interactions is to assess co-occurrence frequencies in observational cancer genomics data. However, this approach is plagued by two major issues. First, the immense number of genes mutated in human cancers, of which the vast majority are mutated in only a small fraction of tumors, makes the study of mutational co-occurrence statistically underpowered for all but the most frequently mutated genes. Second, even when two driver genes co-occur more or less than expected by chance, this can be due to confounding biological factors, such as inactivation of a different gene in the same complex or patient-level selection bias, rather than direct functional epistatic interactions[14]. This is especially true in tumors with a high mutational burden where even mutations in known cancer driver genes may be chance passengers. Thus, a global fitness landscape of tumorigenesis cannot be generated from human data alone, and instead requires direct perturbational experiments and functional genomics approaches[14].

Genetically engineered mouse models are uniquely tractable systems to uncover the phenotypic effects of defined genetic alterations on tumors that develop entirely within their natural in vivo microenvironment[15]. Systems that integrate CRISPR/Cas9-mediated somatic genome editing with conventional genetically engineered mouse models of human cancer have increased the scale at which the consequences of tumor suppressor gene inactivation on autochthonous tumorigenesis can be quantified[16,17]. We recently developed tumor barcode sequencing (Tuba-seq), which integrates barcoded lentiviral-sgRNA/Cre vectors and high-throughput barcode sequencing to uncover the number of neoplastic cells in each tumor of each genotype[18–20].

Here, we initiate and quantify the development of more than one hundred different genotypes of autochthonous lung tumors. This extensive adaptive fitness landscape overlays inactivation of a panel of diverse tumor suppressor genes on top of oncogenic KRAS G12D-, KRAS G12C-, BRAF V600E-, and EGFR L858R-driven lung tumors. *KRAS*, *EGFR*, and *BRAF* are the three most frequently altered oncogenes in lung adenocarcinoma (LUAD)[21] and together drive tumorigenesis in over half of patients (Supplementary Table 1). Their products are canonically depicted in a linear axis—from EGFR to KRAS to BRAF—within the RAS pathway[21–24]. This linear representation implies that the only dimension upon which these oncogenes are functionally different is the quantity of downstream MAPK signaling that they drive. However, each of these oncogenes engages additional pathways, which could generate phenotypic differences between these oncogenes in specific contexts[22–24]. Despite the well-established significance of these RAS pathway oncogenes in lung tumorigenesis, it remains unclear the extent to which these off-axis interactions, even if known phenotypically or biochemically, can drive differential *fitness* effects during tumorigenesis. Mutations within *EGFR*, *KRAS*, and *BRAF* oncogenes are also diverse, and it is unclear whether these mutations are functionally equivalent outside of potential differences in induced RAS signaling.

By generating the most extensive functional survey of oncogene-tumor suppressor interactions to date, we uncover dramatically different tumor suppressive fitness effects across oncogenic contexts, unexpected similarities for oncogenes with strong differences in tumor-driving potential, and surprising effects of off-axis signaling.

## Results

### Oncogenic KRAS G12C is less potent than KRAS G12D in driving lung tumorigenesis

Oncogenic KRAS mutations, predominantly within codon 12, occur in -25% of human LUAD (Supplementary Table 1)[25,26]. To enable CRISPR/Cas9-mediated somatic genome editing in the context of different oncogenic KRAS variants, we generated mice with Cre/lox-regulated alleles of KRAS G12C (*Kras^LSL-G12C*)[27] or KRAS G12D (*Kras^LSL-G12D*)[28] and a Cre/lox-regulated Cas9 allele (*H11^LSL-Cas9*; Fig. 1A)[29]. While the impact of inactivating diverse tumor suppressor genes on KRAS G12D-driven lung cancer growth has been investigated previously[19,20], the functional landscape of tumor suppression within KRAS G12C-driven lung cancer in vivo remains entirely uncharacterized, even though KRAS G12C is the most common oncogenic KRAS variant in human lung cancer (Supplementary Fig. 1). To broadly uncover the genetic interactions between tumor suppressor genes and these oncogenic KRAS variants in vivo, we generated barcoded Lenti-sgRNA/Cre vectors targeting 28 known and putative tumor suppressor genes that are recurrently mutated in human LUAD and represent key cancer pathways (Fig. 1A and Supplementary Fig. 2; "Methods")[21,26,30]. We generated a pool of barcoded Lenti-sgRNA/Cre vectors, which included vectors targeting each of these genes as well as control Lenti-sgRNA/Cre vectors with non-targeting sgRNAs (sg*NT*) and active-cutting sgRNAs (sg*AC*) that target an inert region of the genome (*Rosa26*; sg*R26*) (Lenti-D2G^28-Pool/Cre; Fig. 1A and Supplementary Fig. 2A).

We initiated tumors by delivering Lenti-D2G^28-Pool/Cre to intratracheally intubated *Kras^LSL-G12C;H11^LSL-Cas9* (*G12C;Cas9*) and *Kras^LSL-G12D; H11^LSL-Cas9* (*G12D;Cas9*) mice (Fig. 1A). Given the uncertain oncogenicity of KRAS G12C relative to KRAS G12D, we initiated tumors with several different titers of Lenti-D2G^28-Pool/Cre (from $1.8 \times 10^5$ to $1.35 \times 10^6$ TU (transduction units)/mouse) and analyzed cohorts of mice at 9 and 15 weeks post-tumor initiation (Fig. 1B; $n$ = between 9 and 35 mice per group). At the time of lung collection, *G12C;Cas9* mice had noticeably fewer and smaller surface lung tumors and significantly lower lung weights relative to titer- and timepoint-matched *G12D;Cas9* mice (Fig. 1C and Supplementary Fig. 3A). Histology indicated that *G12C;Cas9* mice had fewer tumors and these appeared smaller than those in *G12D;Cas9* mice (Fig. 1D, E). Lung tumors in both backgrounds were hyperplasias, adenomas, and adenocarcinomas. These results suggest that KRAS G12C is less potent than KRAS G12D in driving lung tumorigenesis, consistent with previous studies using in vivo models of lung and pancreatic cancer[27,31].

### KRAS G12C induces fewer and smaller tumors than KRAS G12D

To quantify the number and size of the tumors in each mouse, and to better understand the dynamics of lung tumor growth driven by these different oncogenic KRAS variants, we performed Tuba-seq on DNA extracted from bulk tumor-bearing lungs from mice across the

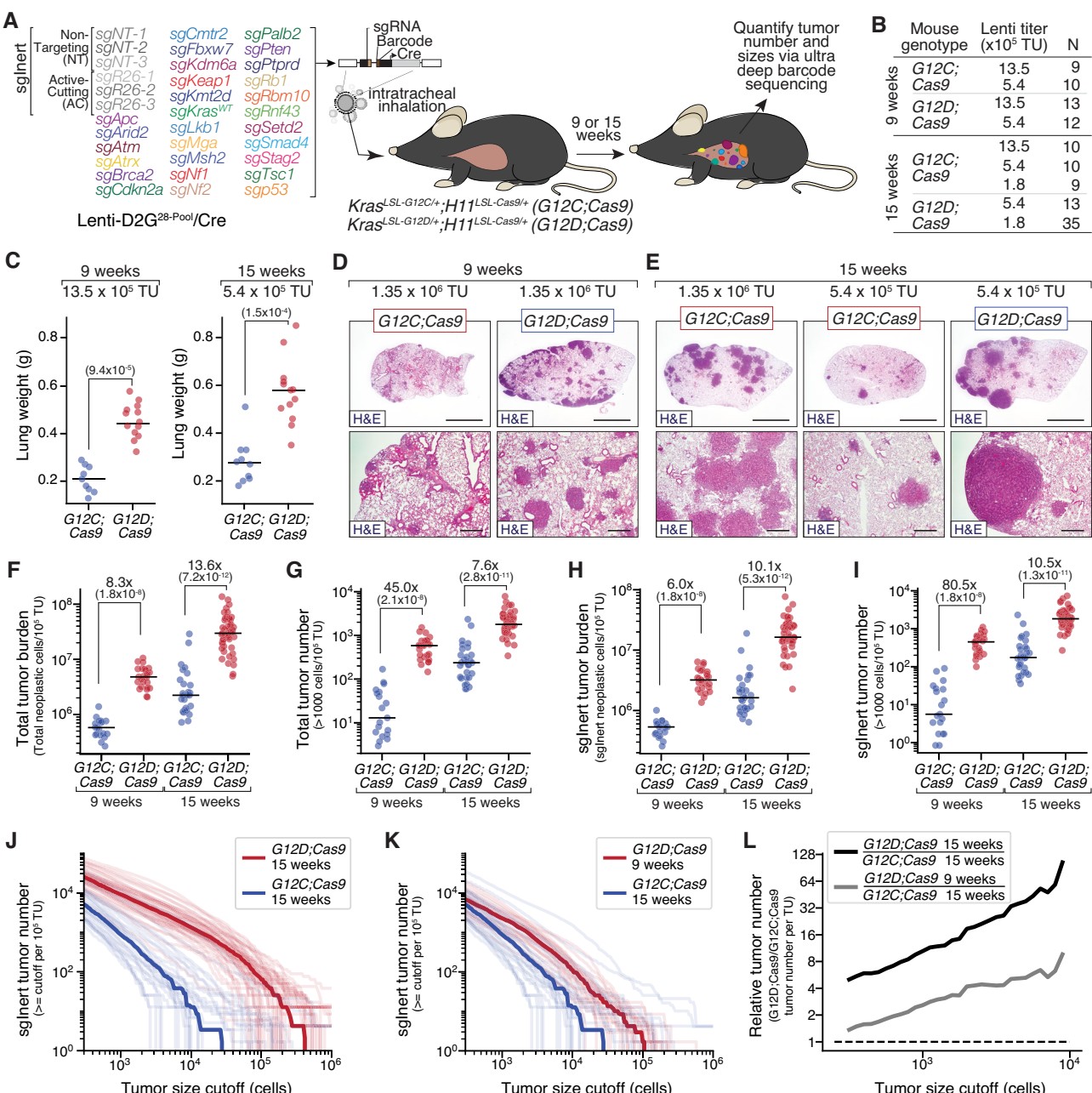

**Fig. 1 | Oncogenic KRAS G12C has a reduced ability to drive initiation and growth of lung tumors in vivo relative to oncogenic KRAS G12D. A** Experimental schematic depicting the composition of the pool of barcoded Lenti-sgRNA/Cre vectors (Lenti-D2G^28-Pool/Cre), mouse genotypes, analysis time points, and readouts. **B** Genotype, time point, lentiviral titer, and number of mice in each group. **C** Lung weights of mice transduced with the indicated titers of Lenti-D2G^28-Pool/Cre. Genotype and time post-tumor initiation are indicated. Each dot represents a mouse, and the bar is the median. Fold difference between medians and significance calculated using a two-sided Wilcoxon rank-sum test (*P* values = number in parentheses) are shown. *n* = 22 biologically independent animals used to calculate significance in each plot. **D, E** Representative histology of lungs from mice (from *n* = 3 mice for each timepoint-genotype pair). Mouse genotype, virus titer delivered to each mouse, and time post-tumor initiation are shown. Top scale bars = 3 mm; bottom scale bars = 500 μM. **F–I** Total number of neoplastic cells (**F**) and total number of tumors greater than 1000 cells in size (**G**) across all Lenti-sgRNA/Cre vectors,

normalized to viral titer. Total number of neoplastic cells (**H**) and total number of tumors greater than 1000 cells in size (**I**) only for Lenti-sgInert/Cre vectors (tumors driven by oncogenic Kras alone), normalized to viral titer. Mouse genotypes and time points are indicated. Each dot represents a mouse, and the bar is the median. Fold difference and significance calculated using a two-sided Wilcoxon rank-sum test (*P* values = number in parentheses) are shown. *n* = 44 (9-week comparison) and *n* = 77 (15-week comparison) biologically independent animals were used to calculate significance. **J, K** Number of tumors at or above the tumor size cutoff in *G12D;Cas9* mice at 15 weeks and *G12C;Cas9* mice at 15 weeks (**J**) or *G12D;Cas9* at 9 weeks and *G12C;Cas9* at 15 weeks (**K**) post-tumor initiation. Each transparent line represents a mouse, and the solid line is the median tumor number. **L** Fold change in median tumor number between *G12D;Cas9* and *G12C;Cas9* at 15 weeks (black line) and *G12D;Cas9* at 9 weeks versus *G12C;Cas9* at 15 weeks (gray line) post-tumor initiation.

different titers and timepoints (Supplementary Fig. 2A). Tuba-seq accurately quantifies the number of neoplastic cells in each tumor (cells directly descending from the tumor cell of origin) through deep sequencing of the multi-component barcode encoded within each genomically integrated lentivirus (Supplementary Fig. 2A–C)[19,20]. This allowed us to estimate the total tumor burden (total neoplastic cells/TU) and total tumor number (number of clonal barcoded tumors with >1000 neoplastic cells/TU) in each mouse (Supplementary Fig. 2A–C; "Methods")[19,20]. Nine weeks post-tumor initiation, total tumor burden (normalized to titer) was >eightfold lower in *G12C;Cas9* mice than in *G12D;Cas9* mice (Fig. 1F and Supplementary Fig. 3B; $P < 2 \times 10^{-8}$). This difference was slightly greater and still highly significant 15 weeks post-tumor initiation (Fig. 1F and Supplementary Fig. 3C; $P < 7 \times 10^{-12}$). At both 9 and 15 weeks post-tumor initiation, *G12C;Cas9* mice also had many fewer tumors per TU than *G12D;Cas9* mice (Fig. 1G and Supplementary Fig. 3B, C; $P < 2 \times 10^{-8}$). Tumor number increased linearly with titer, consistent with a lack of inter-tumor competition even at high tumor burden. Thus, when considering all tumors independently of their engineered tumor suppressor inactivation, KRAS G12C drives substantially less neoplastic growth than KRAS G12D.

Lenti-sg*NT*/Cre and Lenti-sg*R26*/Cre (sgInert) vectors induce the expression of oncogenic KRAS from the engineered alleles without CRISPR/Cas9-mediated inactivation of any gene, generating tumors driven solely by oncogenic KRAS. Before exploring the impact of inactivation of each tumor suppressor gene on tumor growth, we restricted our analysis to these sgInert "KRAS-only" tumors. sgInert tumor burden and tumor number were also dramatically lower in *G12C;Cas9* mice relative to *G12D;Cas9* mice, at both 9 and 15 weeks post-tumor initiation (Fig. 1H, I and Supplementary Fig. 3B–D; all $P < 10^{-7}$). As expected, for each oncogene and timepoint, sgInert tumor burden (normalized for titer) is lower than total tumor burden, which reflects the fact that the total tumor burden includes tumors in which tumor suppressor genes have been inactivated (Fig. 1F, H). For example, at 15 weeks, the ratio of the median total tumor burden to median sgInert tumor burden (normalized to their relative viral titers) was 1.85 ($P = 6.4 \times 10^{-4}$, Wilcoxon rank-sum test) in *G12D;Cas9* mice and 1.37 ($P = 0.17$) in *G12C;Cas9* mice.

To further investigate the different abilities of oncogenic KRAS G12C and KRAS G12D to initiate lung tumors and drive their growth, we explored the distribution of sgInert tumor sizes (Fig. 1J–L and Supplementary Fig. 3E–G). A comparison of the two models 15 weeks post-tumor initiation revealed fewer KRAS G12C tumors than KRAS G12D tumors above any minimum size cutoff (Fig. 1J). Furthermore, the KRAS G12D tumor size distribution had a longer tail of large tumors, suggesting that its increased tumor number might be driven by more rapid growth than tumors driven by KRAS G12C (Fig. 1J). In support of this notion, the shape of the KRAS G12C tumor size distribution 15 weeks post-tumor initiation was similar to that of the KRAS G12D tumor size distribution at the earlier 9-week timepoint (Fig. 1K). However, while the shapes of the distributions at these two timepoints were quite well matched (Fig. 1K), KRAS G12D consistently produced ~2–4× greater tumor number than KRAS G12C, suggesting that KRAS G12D may also drive greater levels of tumor initiation (Fig. 1L). These results are all consistent with a model in which KRAS G12C is less potent at initiating lung tumors and less able to drive the expansion of established tumors in vivo than KRAS G12D.

**Diverse tumor suppressor genes have strikingly similar effects on the initiation and growth of KRAS G12C- and KRAS G12D-driven lung tumors**

Having used Tuba-seq to uncover differences in the baseline ability of KRAS G12C and KRAS G12D to initiate lung tumors and drive their growth, we next analyzed the impact of inactivating each of the 28 putative tumor suppressor genes on the growth of lung tumors driven by these oncogenes (Fig. 2A, B and Supplementary Fig. 4A). To

compare the effects of inactivating each targeted gene across onco-genes, we analyzed all tumors above oncogene-specific tumor size (number of neoplastic cells) cutoffs that matched the number of sgInert tumors in each oncogenic context ("Methods"). Matching cutoffs in this way allowed us to account for differential oncogene-intrinsic growth dynamics. We used a minimum tumor size cutoff of 1600 cells for *G12D;Cas9* mice at 15 weeks post-tumor initiation, 600 cells for *G12D;Cas9* mice at 9 weeks, 400 cells for *G12C;Cas9* mice at 15 weeks, and 300 cells for *G12C;Cas9* at 9 weeks. We then compared the sizes of tumors in which each tumor suppressor gene was targeted to the sizes of sgInert tumors.

Inactivation of many of these genes led to the development of larger KRAS G12C- and KRAS G12D-driven tumors (Fig. 2A, B). These tumor suppressive effects were highly reproducible across eleven *G12D;Cas9* study groups (pre-defined cohorts of mice of identical genotype, administered viral titer, date of tumor initiation, and date of take down) and four *G12C;Cas9* study groups (Fig. 2C and Supplementary Fig. 5; data from 243 *G12D;Cas9* mice and 47 *G12C;Cas9* mice; Pearson $r \geq 0.95$ and $r \geq 0.87$, respectively, for each comparison). To further assess whether these effects were driven by on-target cutting, we generated a second barcoded Lenti-sgRNA vector targeting each gene with a distinct sgRNA. We initiated tumors using a new virus pool that contained the original and new sgRNAs targeting each gene, and compared the tumor suppressive effects measured using each of the two sgRNAs per gene. The 95th percentile relative tumor size ratios were highly correlated between the two sgRNAs (Supplementary Fig. S5D; $r = 0.93$). Tumor suppressive effects were also correlated when using vectors with the same sgRNA but on different plasmid backbones (Supplementary Fig. S5E; $r = 0.96$).

Comparing size distributions of tumors of each tumor suppressor genotype revealed consistent effects between KRAS G12C- and KRAS G12D-driven lung tumors. Despite the differences in oncogenic potential of KRAS G12C and KRAS G12D, the tumor suppressive effects at 15 weeks post-initiation were highly correlated (Fig. 2A, B, D; Spearman $\rho = 0.92$). Tumor suppressive effects were also highly correlated between KRAS G12C-driven tumors at 15 weeks post-tumor initiation and KRAS G12D-driven tumors at 9 weeks post-tumor initiation, when sgInert tumors were most similar in size (Fig. 2E; Spearman $\rho = 0.90$). In fact, across all comparisons of timepoints and oncogenic alleles, tumor suppressive effects were well correlated (Fig. 2D, E and Supplementary Fig. 6; all Spearman and Pearson correlations $\rho \geq 0.88$).

Our pool contained Lenti-sgRNA/Cre vectors targeting two established negative regulators of oncogenic KRAS signaling: the NF1 RAS GTPase-activating protein and wild-type KRAS (*Kras^WT^*). Inactivation of either *Nf1* or *Kras^WT^* has been shown to increase downstream signaling and enable faster growth of oncogenic KRAS G12D-driven lung tumors in vivo[19,32,33]. Contrary to our expectation that inactivation of these negative regulators would have a greater effect on tumors driven by the weaker KRAS G12C variant, inactivation of *Nf1* or *Kras^WT^* increased KRAS G12C- and KRAS G12D-driven tumor growth to the same extent. This result was consistent across study groups in multiple independent studies (Fig. 2F, G). Thus, despite the large difference in the ability of KRAS G12C and KRAS G12D to drive tumor growth, as well as their known biochemical differences[34], lung tumor growth driven by these oncogenic variants was similarly affected by alterations in oncogene-proximal tumor suppressors.

While the impact of inactivating most tumor suppressor genes was similar in lung tumors driven by KRAS G12C and KRAS G12D, there were some notable exceptions. Inactivation of the H3K4 mono- and di-methyltransferase *Kmt2d* or the cap-specific mRNA methyltransferase *Cmtr2* consistently increased the growth of KRAS G12C-driven tumors less than KRAS G12D-driven tumors (Fig. 2H, I; $P = 1.8 \times 10^{-6}$, $2.6 \times 10^{-24}$, respectively). Collectively, these results are inconsistent with the model of diminishing-returns epistasis where adaptive mutations are

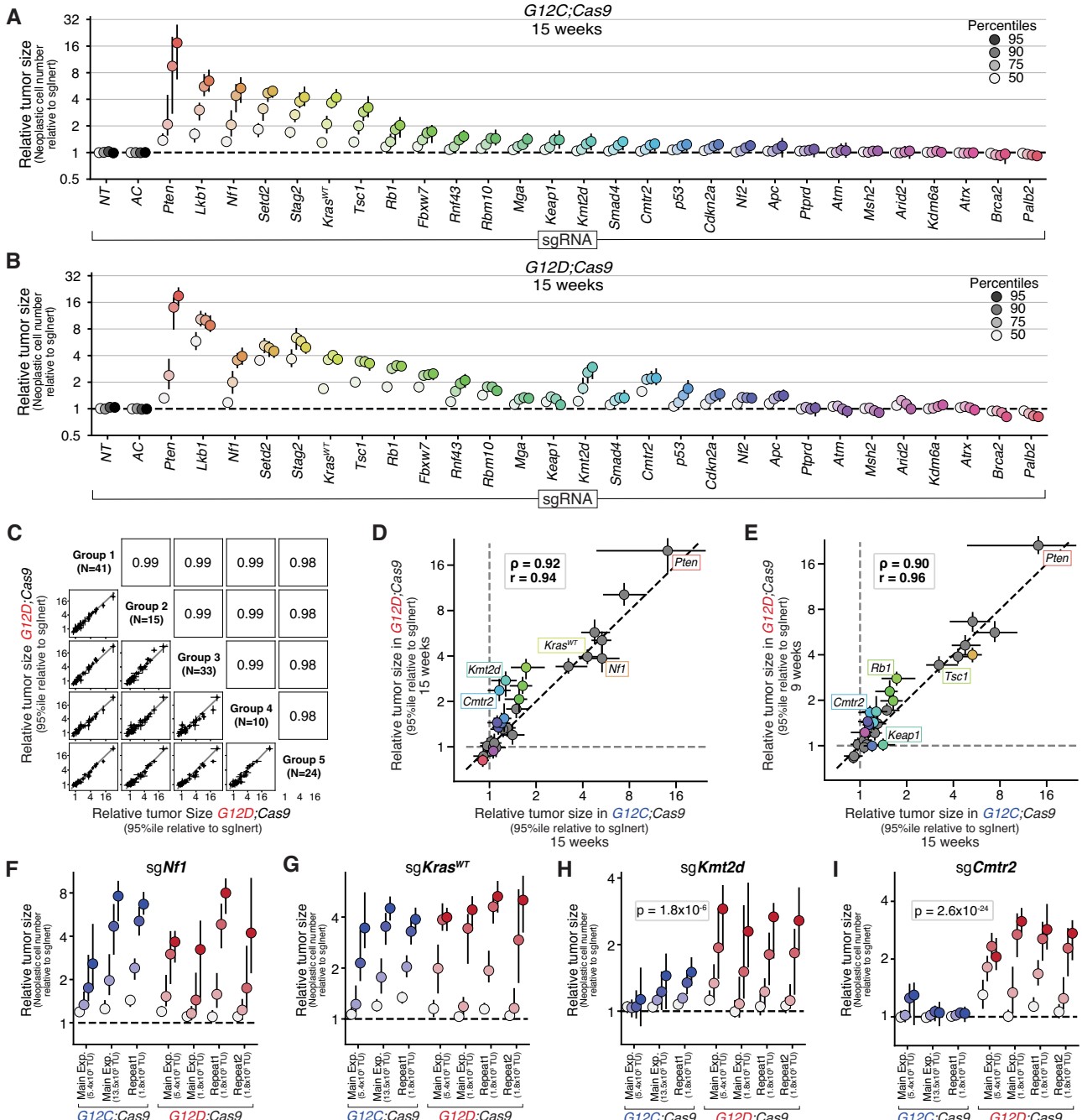

**Fig. 2 | Tumor suppressor genes have strikingly similar effects on the initiation and growth of KRAS G12C- and G12D-driven lung tumors. A**, **B** Relative size (neoplastic cells) of the tumor at the indicated percentiles of the tumor size distributions for barcoded Lenti-sgRNA/Cre vectors targeting each gene, relative to the size of the sgInert tumor at the same percentile, in *G12C;Cas9* mice (*n* = 29 biologically independent animals) (**A**) and *G12D;Cas9* mice (*n* = 48 biologically independent animals) (**B**) at 15 weeks post-tumor initiation. The point represents the value calculated from the initial data, and 95% confidence intervals from bootstrapping are shown. **C** 95th percentile relative tumor sizes (relative to sgInert) for 5 of the *G12D;Cas9* study groups (see Supplementary Fig. 5A, B for comparisons between additional study groups). Each point represents the tumors initiated with one Lenti-sgRNA/Cre vector and the bars are the 95th percent confidence intervals determined by bootstrapping. Gray line indicates equal effect. Pearson r is indicated. **D**, **E** Relative size of the tumor at the 95th percentile of the tumor size distributions in *G12D;Cas9* mice at 15 weeks (**D**) or *G12D;Cas9* mice at 9 weeks (**E**) (*n* = 25 biologically independent animals) versus in *G12C;Cas9* mice at 15 weeks post-tumor initiation. Each dot represents the tumors initiated from one Lenti-

sgRNA/Cre vector and the bars are the 95th percent confidence intervals. Genes where the 95% CI excluded no effect in *G12C;Cas9* and *G12D;Cas9* mice are shown in color and some key genes are labeled. The black dotted line indicates equal effect. Spearman rank-order correlation (ρ) and Pearson correlation (*r*) are indicated. **F**–**I** Relative size of the tumor at the indicated percentiles (see legend in (**A**, **B**)) of the tumor size distributions for barcoded Lenti-sgRNA/Cre vectors targeting *Nf1* (**F**), *Kras^{WT}* (**G**), *Kmt2d* (**H**), and *Cmtr2* (**I**) across multiple arms of our main experiment and repeat studies in *G12C;Cas9* and *G12D;Cas9* mice. The significance of oncogene differences at the 95th percentile were calculated by combining the study groups for each oncogene using inverse variance weighting and comparing the resulting means and variances under a normally distributed null. Bonferroni-corrected, one-sided *P* values are shown for significant genes. For each study group from left to right along the *x* axes, data include *n* = 10, *n* = 10, *n* = 18, *n* = 13, *n* = 35, *n* = 23, and *n* = 33 biologically independent animals. Note that the two study groups from repeat studies in *G12D;Cas9* correspond to Group 8 and Group 3 in Supplementary Fig. 5.

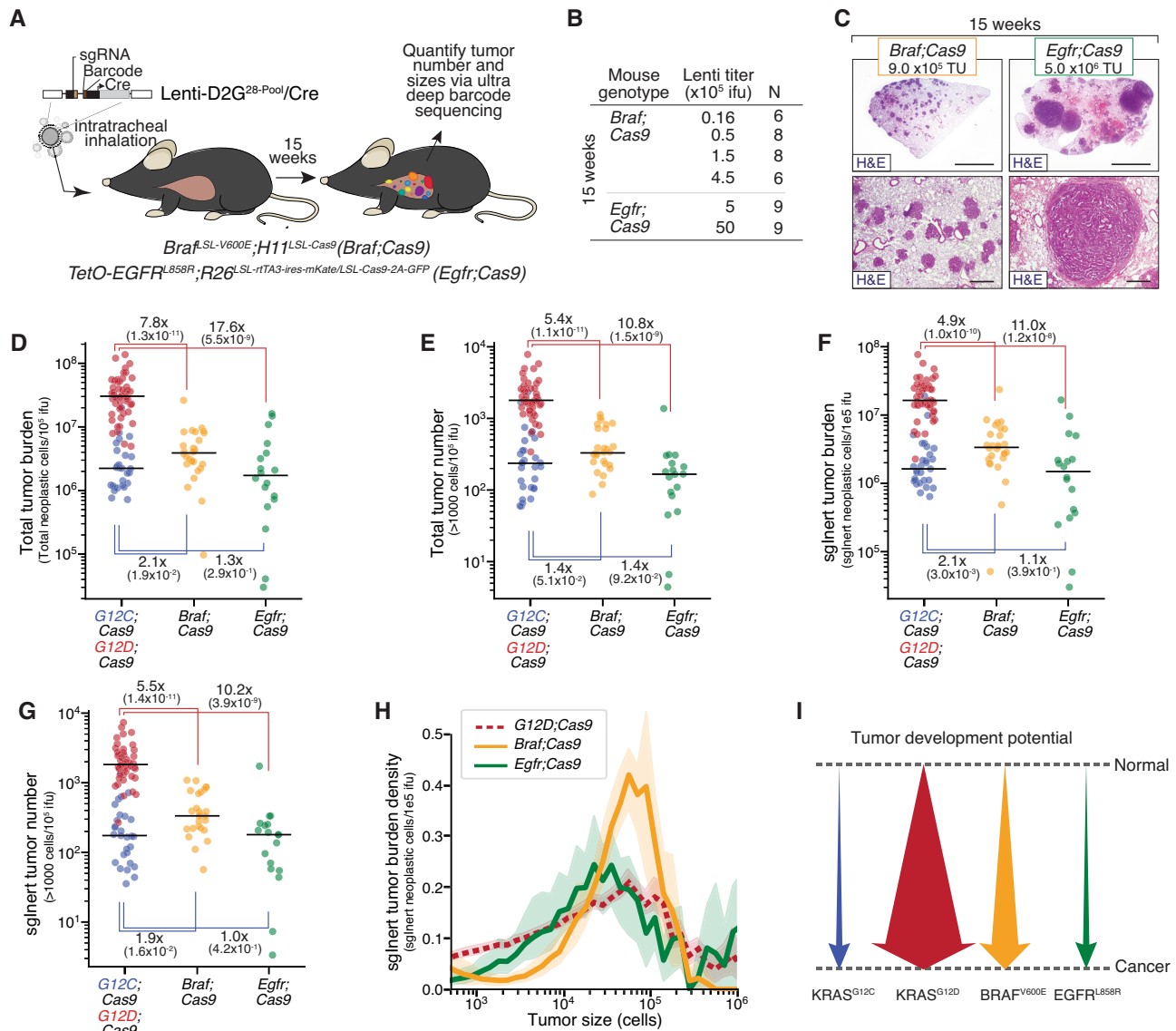

**Fig. 3 | Oncogenic BRAF, EGFR, and KRAS have different abilities to initiate lung tumorigenesis and drive tumor growth. A** Experimental schematic showing the design of barcoded Lenti-sgRNA/Cre vectors (Lenti-D2G[28-Pool]/Cre), mouse genotypes, and analysis timepoints. **B** Mouse genotype, time point, lentiviral titer, and number of mice in each experimental group. **C** Representative histology of lungs from mice. Mouse genotype, viral titer, and time point post-tumor initiation are shown. Top scale bars = 3 mm; bottom scale bars = 500 μM. Titer represented in the *Braf;Cas9* image was 900,000 TU, from a mouse in a separate titering experiment that used a similar virus pool. Titer represented in the *Egfr;Cas9* image was 5,000,000 TU. **D**–**G** Total number of neoplastic cells (**D**) and total number of tumors greater than 1000 cells in size (**E**) across all Lenti-sgRNA/Cre vectors, normalized to viral titer. Total number of neoplastic tumors cells (**F**) and total number of tumors greater than 1000 cells in size (**G**) only for Lenti-sgInert/Cre vectors (tumors driven by oncogene alone), normalized to viral titer. Mouse genotypes are

indicated. Each dot represents a mouse, and the bar is the median. Fold differences between medians and significance calculated using a two-sided Wilcoxon rank-sum test (*P* values = number in parentheses) are shown. Fold differences are ratios of the following pairs, moving clockwise from the upper left: *G12D;Cas9/Braf;Cas9, G12D;Cas9/Egfr;Cas9, G12C;Cas9/Egfr;Cas9, Braf;Cas9/G12C;Cas9.* **H** The density function of sgInert tumor burden as a function of log(tumor size) at 15 weeks for *G12D;Cas9, Braf;Cas9,* and *Egfr;Cas9.* Comparison to *G12C;Cas9* can be found in Supplementary Fig. 3F, G. Error bands represent the 95% confidence interval determined from bootstrapping. For all panels, *Egfr;Cas9* mice are represented by *n* = 18 biologically independent animals, *Braf;Cas9* mice are represented by *n* = 28 biologically independent animals, *G12D;Cas9* mice are represented by *n* = 48 biologically independent animals, and *G12C;Cas9* mice are represented by *n* = 29 biologically independent animals. **I** Schematic representation of the ability of each indicated oncogenic allele to drive in vivo lung tumor formation.

expected to provide greater fitness benefit on the less fit genetic background—in this case, KRAS G12C.

## Oncogenic BRAF and EGFR have distinct tumor-initiating and growth-promoting abilities

Oncogenic mutations in BRAF and EGFR occur frequently in human LUAD, underscoring the importance of activation of the RAS pathway in a large fraction of these tumors (Supplementary Table 1)[21,26]. Oncogenic BRAF mutations (including those at the hotspot V600)

occur in ~6% of LUAD, while oncogenic EGFR mutations occur in ~27% (AACR Project GENIE) of LUAD (Supplementary Table 1)[21,26]. BRAF- and EGFR-driven lung cancers have been modeled in mice using a Cre/lox-regulated conditionally activatable allele of BRAF V600E (*Braf*[CA-V600E])[35] and a doxycycline regulated *EGFR*[L858R] transgene[36,37]. To quantify the ability of oncogenic BRAF and EGFR to initiate lung tumors and drive their expansion in vivo, as well as to uncover whether tumor suppressor effects are consistent across these oncogenic contexts in lung cancer, we initiated tumors with Lenti-D2G[28-Pool]/Cre in *Braf*[CA-V600E];

$H11^{LSL-Cas9}$ (*Braf;Cas9*) and *tetO-EGFR*$^{L858R}$;*Rosa26*$^{LSL-rtTA3-ires-mKate/LSL-Cas9-2a-GFP}$ (*Egfr;Cas9*) mice (Fig. 3A)[36]. Mice received several different titers of Lenti-D2G$^{28-Pool}$/Cre (from $1.6 \times 10^4$ to $5 \times 10^6$ TU/mouse) and were analyzed 15 weeks post-tumor initiation (Fig. 3B). Consistent with previous observations, *Braf;Cas9* mice developed lung tumors that appeared more uniform in size than oncogenic KRAS-driven or oncogenic EGFR-driven tumors (Figs. 1D, E and 3C; Lung weights shown in Supplementary Fig. 7A, B)[35,38]. BRAF and EGFR-driven tumors were adenomas and adenocarcinomas (Fig. 3C).

Tuba-seq analysis of DNA extracted from bulk tumor-bearing lungs allowed us to quantify tumor burden and size across mouse genotypes, and thus determine the tumorigenic potential of BRAF V600E and EGFR L858R relative to KRAS G12C and KRAS G12D. Total tumor burden and total tumor number in *Braf;Cas9* mice was higher than in *G12C;Cas9* mice but >fivefold lower than in *G12D;Cas9* mice (Fig. 3D, E; $P = 1.9 \times 10^{-2}$, $P = 1.3 \times 10^{-11}$). *Egfr;Cas9* mice had slightly lower tumor burden and number than *Braf;Cas9* mice and >tenfold fewer tumors than *G12D;Cas9* mice (Fig. 3D, E; $P = 1.5 \times 10^{-9}$). These results remained consistent after restricting the analysis to sgInert-containing tumors driven by oncogenic BRAF or EGFR (Fig. 3F, G; $P = 1.6 \times 10^{-2}$).

Interestingly, the distribution of BRAF-driven tumor sizes was strikingly different from that of other oncogenic contexts. Exceptionally large tumors accounted for a much smaller percentage of the total tumor burden in *Braf;Cas9* mice than in any other mouse genotype: only 0.9% of neoplastic cells in *Braf;Cas9* mice were from tumors with >300,000 cells compared with 13.1% and 25.4% for *G12D;Cas9* and *Egfr;Cas9* mice, respectively. This is consistent with previous reports that BRAF tumors hit a maximum size threshold and stop growing[38]. However, unlike in *G12C;Cas9*, *G12D;Cas9*, and *Egfr;Cas9* mice, a majority of the total tumor burden in *Braf;Cas9* mice was from tumors with 30,000–300,000 neoplastic cells. (62.2% in *Braf;Cas9* mice compared with 36.0% and 32.3% in *G12D;Cas9* and *Egfr;Cas9* mice, respectively) (Fig. 3H and Supplementary Fig. 7C, D). In comparison to *Braf;Cas9* mice, a much greater fraction of the total neoplastic burden in *G12D;Cas9* mice arose from smaller tumors. The difficulty in quantifying these smaller tumors using histological methods might explain why previous studies have suggested that BRAF V600E is a stronger driver of lung tumorigenesis than KRAS G12D[39,40]. Collectively, these results indicate that different oncogenes in the EGFR/KRAS/BRAF axis have dramatically different effects on tumor initiation and growth (Fig. 3I).

## Oncogenic BRAF and EGFR redefine the landscape of tumor growth suppression

We next investigated the impact of tumor suppressor gene inactivation on the growth of BRAF V600E- and EGFR L858R-driven lung tumors and compared tumor suppressor effects across all four oncogenic alleles. Very few coincident tumor suppressor alterations have been investigated in the context of oncogenic BRAF-driven autochthonous lung tumors, and the extent to which tumor suppressor effects differ in EGFR-driven tumors remains poorly understood (Supplementary Fig. 1)[36,41–44]. Interestingly, the overall tumor suppressive landscapes of BRAF- and EGFR-driven lung tumors were dramatically different from each other as well as from oncogenic KRAS-driven tumors (Figs. 2A, B and 4A, B). Indeed, the effects of tumor suppressor inactivation on growth across oncogenic KRAS-, BRAF-, and EGFR-driven tumors were uncorrelated (Fig. 4C, D; Spearman $\rho = 0.41$ for BRAF versus G12C, $\rho = 0.14$ for EGFR versus G12C). While inactivation of some tumor suppressor genes increased growth across all contexts (e.g., *Pten*), those were the exception (Figs. 2 and 4 and Supplementary Fig. 8). Importantly, several tumor suppressor genes impacted tumorigenesis as anticipated. Inactivation of *Nf1* increased size of KRAS- and EGFR-driven tumors while having no effect on BRAF-driven tumors (Supplementary Fig. 8D). Inactivation of *Kras*$^{WT}$ increased the

growth of KRAS G12C- and KRAS G12D-driven tumors but had no effect of BRAF-driven tumors and reduced the growth of EGFR-driven tumors (consistent with KRAS being an important downstream effector) (Fig. 4E).

Inactivation of many tumor suppressor genes had strikingly different effects on the growth of BRAF-driven tumors compared to tumors driven by either KRAS variant (Fig. 4C and Supplementary Fig. 8A). While the tumor suppressive effects of inactivating *Pten*, *Rnf43*, and *Apc* are consistent with previous data on these genes and/or related pathways in BRAF-driven lung cancer[38,41–43,45–48], the general decreases in magnitude relative to oncogenic KRAS were unexpected (Fig. 4F and Supplementary Fig. 8E). The effect of coincident tumor suppressor inactivation could generally be reduced due to diminishing-returns epistasis in fast-growing BRAF V600E-driven tumors. However, lower effect magnitudes were not universal as inactivation of *Rnf43* or *Fbxw7* increased tumor growth as much or more in *Braf;Cas9* mice than in *G12C;Cas9* mice (Fig. 4F, G). Thus, the impact of certain tumor suppressor pathways on tumor growth largely depends on the oncogenic context.

The differences between tumor suppressive effects in EGFR-driven lung cancer and the other oncogenic contexts were even more pronounced (Fig. 4B, D and Supplementary Fig. 8B, C). Inactivation of many genes that were functional tumor suppressors in KRAS-driven lung tumors, including *Lkb1*, *Setd2*, and *Kmt2d*, were deleterious in EGFR-driven lung tumors (Fig. 4B, D and Supplementary Fig. 8B, G–I). Conversely, inactivation of *p53* increased the overall growth of EGFR-driven lung tumors more than in any other oncogenic context (Supplementary Fig. 8F). These differences represent the clearest indication of a rugged landscape of oncogene-tumor suppressor interactions; whether a second step (tumor suppressor inactivation) led uphill or downhill depended strongly on which first uphill step was taken (EGFR or KRAS or BRAF).

## Genetic interactions between oncogenes and tumor suppressors impact the earliest stages of tumor development

The initiation of tumors using the same virus pool in mice with and without the *Cas9* allele enables quantification of the impact of each gene on tumor number, as the relative numbers of tumors in mice without the *Cas9* allele constitute an in vivo titering experiment that reveals the representation of each virus in the pool (Supplementary Fig. 9)[19]. As anticipated, mice with conditional oncogene alleles but lacking the $H11^{LSL-Cas9}$ allele (Cas9-negative mice) transduced with Lenti-D2G$^{28-Pool}$/Cre had much lower overall tumor burden than their Cas9-positive counterparts, and no Lenti-sgRNA/Cre vector had any effect on tumor sizes (Supplementary Fig. 4B, C shows the tumor size percentiles for the corresponding Cas9-positive cohort). Furthermore, the *Cas9* allele had little-to-no effect on tumor growth (Supplementary Fig. 4D; see "Methods").

As the Cas9-negative cohorts behaved as expected, each Cas9-positive oncogenic context was compared to a corresponding cohort of Cas9-negative mice initiated with the same virus pool, allowing us to quantify the impact of each tumor suppressor gene on tumor initiation/early tumor expansion (Fig. 5A–D and Supplementary Fig. 9B–E). To capture the signal most representative of tumor initiation, we used the lowest possible cutoff for all oncogenes, which was 500 neoplastic cells given the resolution of our method. Although this approach did not control for differences in baseline oncogenic potency as was done for tumor size enrichment, we did not see any systematic bias of the tumor number enrichment when comparing pairs of oncogenes.

As was the case for tumor growth effects, inactivation of most genes had similar effects on the number of KRAS G12C- and KRAS G12D-driven tumors (Fig. 5A, B and Supplementary Fig. 10A). However, *Cdkn2a* inactivation increased tumor number in the KRAS G12D context more than in KRAS G12C, consistent with genomic analyses of human non-small cell lung cancers describing enrichment of *CDKN2A*

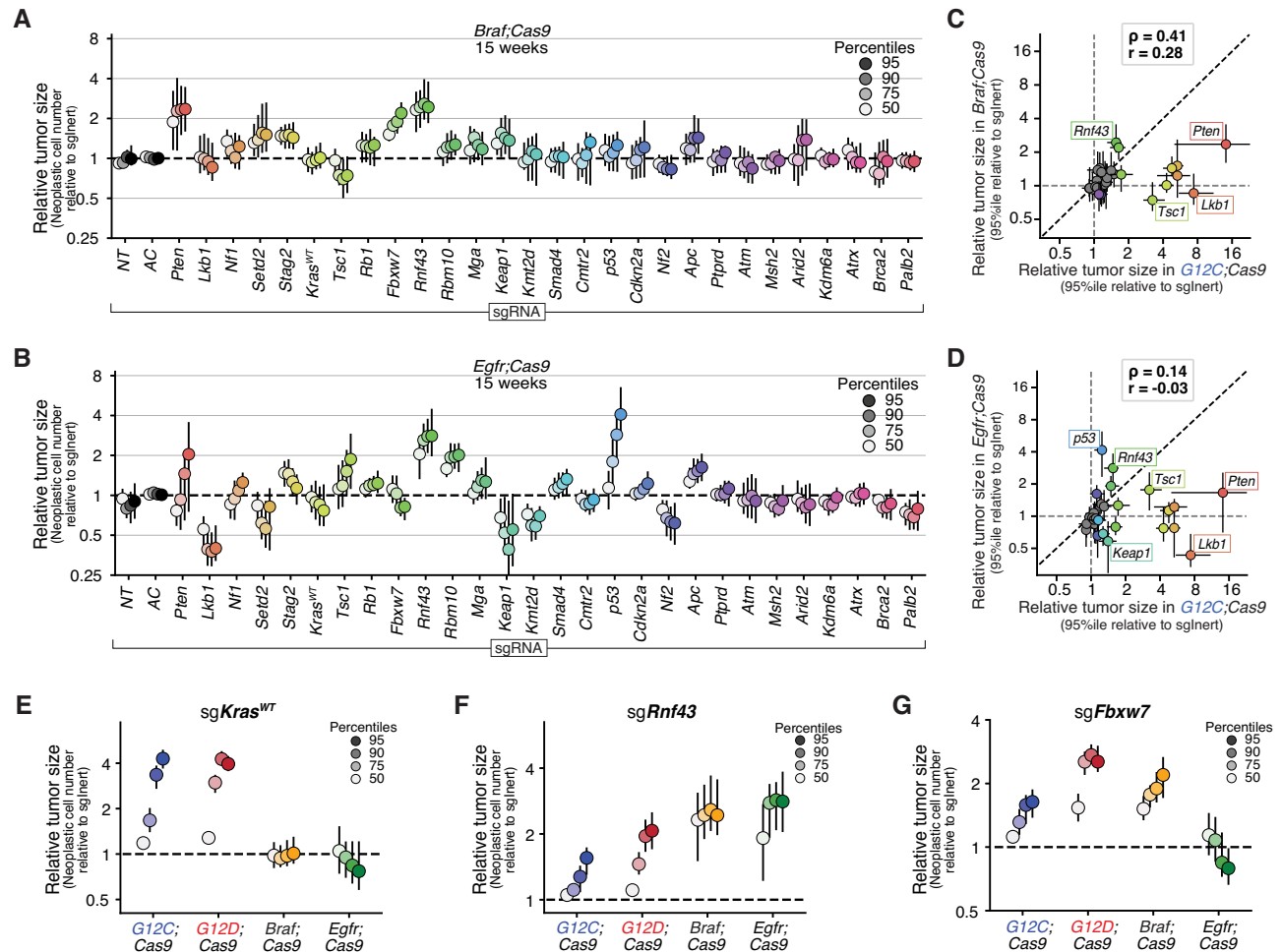

**Fig. 4 | Oncogenic driver defines the landscape of tumor growth suppression in lung cancer. A, B** Relative size (neoplastic cells) of the tumor at the indicated percentiles of the tumor size distributions for barcoded Lenti-sgRNA/Cre vectors targeting each gene, relative to the size of the sgInert tumor at the same percentile, in *Braf;Cas9* mice (**A**) and *Egfr;Cas9* mice (**B**) at 15 weeks post-tumor initiation. Each dot represents the relative tumor size of tumors initiated from one Lenti-sgRNA/Cre vector at a given percentile and the bars are the 95% confidence intervals.
**C, D** Relative size of the tumor at the 95th percentile of the tumor size distributions in *G12C;Cas9* versus *Braf;Cas9* mice (**C**) and *Egfr;Cas9* mice (**D**). Each dot represents the tumors initiated from one Lenti-sgRNA/Cre vector and the bars are the 95% confidence intervals. Genes where the 95% confidence interval excluded no effect

are shown in color and some key genes are labeled. Black dotted line indicates equal effect. Spearman rank-order correlation (ρ) and Pearson correlation (*r*) are indicated. **E–G** Relative size of the tumor at the indicated percentiles of the tumor size distributions for Lenti-sgRNA/Cre vectors targeting *Kras^{WT}* (**E**), *Rnf43* (**F**), and *Fbxw7* (**G**) and in tumors in the indicated genotypes of mice. Each dot represents the tumors initiated from one Lenti-sgRNA/Cre vector at a given percentile and the bars are the 95% confidence intervals. For all panels in this figure, *Egfr;Cas9* mice are represented by *n* = 18 biologically independent animals, *Braf;Cas9* mice are represented by *n* = 28 biologically independent animals, *G12D;Cas9* mice are represented by *n* = 48 biologically independent animals, and *G12C;Cas9* mice are represented by *n* = 29 biologically independent animals.

mutations in tumor with *KRAS G12D* relative to *KRAS G12C* or other KRAS variants[30,49,50]. Furthermore, the impact of different tumor suppressors on tumor number varied across oncogenic KRAS, BRAF, and EGFR contexts (Fig. 5 and Supplementary Fig. 10). Interestingly, in *Braf;Cas9* mice, inactivation of tumor suppressors had little effect on tumor number (Fig. 5C). Conversely, the number of EGFR-driven tumors was greatly impacted by coincident tumor suppressor inactivation. These effects were large in magnitude (e.g., >sixfold increase for sg*Pten*) and included many genes that reduce tumor number (e.g., >fourfold decrease for sg*Lkb1*), suggesting several of these tumor suppressor genes do not in fact suppress EGFR-driven tumors (Fig. 5D). Thus, much like the effects on tumor growth, tumor initiation/early expansion is highly context-dependent with complex and diverse genetic interactions influencing even the earliest steps of lung carcinogenesis.

Finally, as was found before in the KRAS G12D context[19], across all four oncogenic contexts in our study the impact of inactivating tumor suppressor genes on tumor initiation/early expansion and

tumor growth did not correlate (Fig. 5E–H). This suggests that the genes and pathways that regulate the earliest stages of tumorigenesis are largely non-overlapping with those that modulate later tumor growth.

**In vivo tumor suppressive effects correlate with the frequency of tumor suppressor alterations in human tumors when the burden of passenger mutations is low**

We next performed a retrospective analysis to determine whether the frequencies of tumor suppressor alterations in human lung adenocarcinoma (2204 samples from AACR Project GENIE) correlate with the fitness effects elucidated using our in vivo models. Such a correlation is expected if the mutation frequencies are driven by effects on tumor growth that our model recapitulates, but may be undermined by the fact that (i) tumor suppressor genes could have complex epistatic relationships with each other; for instance, the inactivation of a gene, complex, or pathway can make the inactivation of another gene in the same complex or pathway functionally redundant and thus neutral,

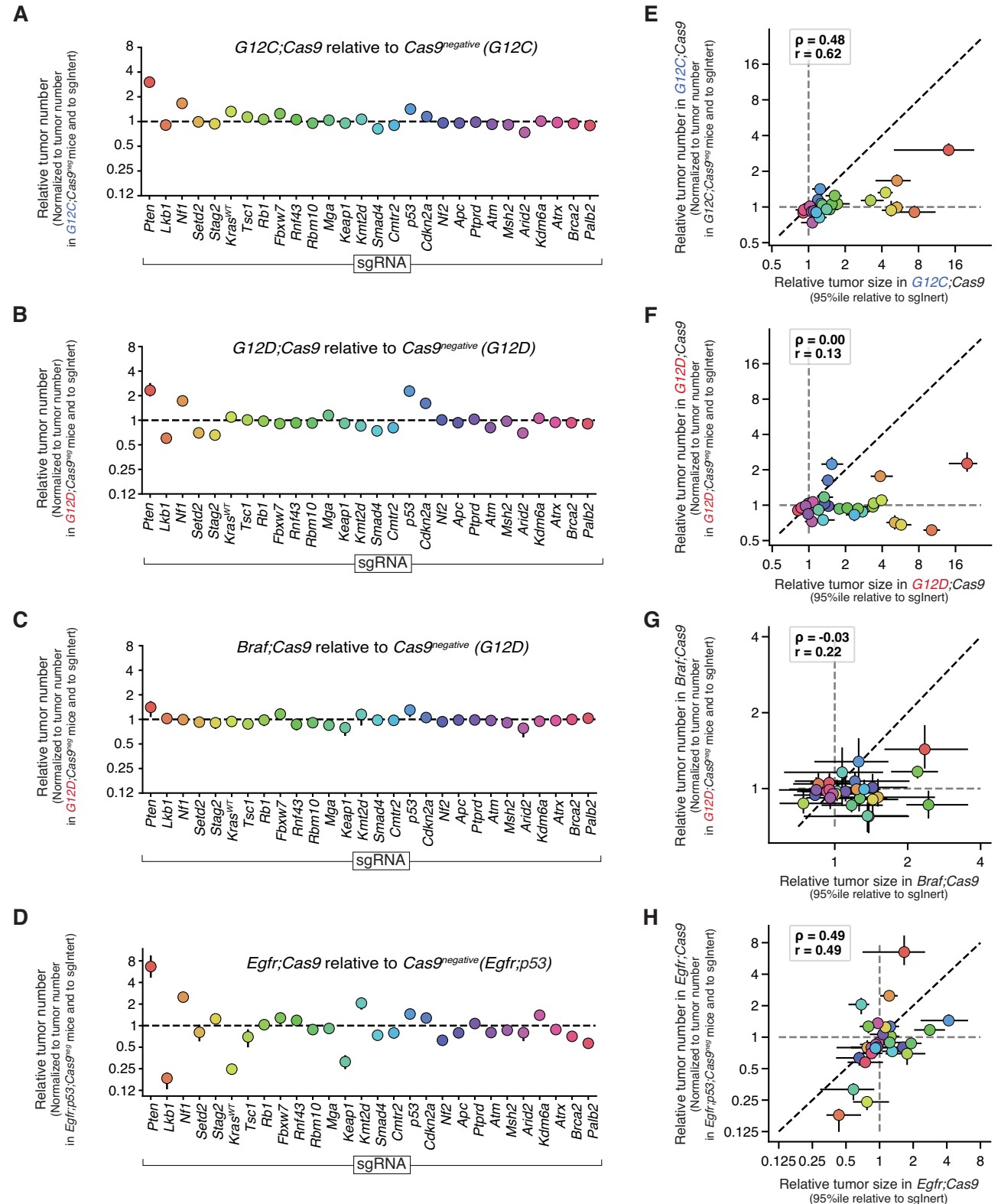

**Fig. 5 | The impact of different tumor suppressors on lung tumor number is dependent on oncogenic context and largely independent of effects on tumor growth. A–D** Impact of inactivating each gene on relative tumor number in the indicated genotypes of mice. Error bars represent the 95% confidence interval determined by bootstrapping the tumors and mice. **E–H** Relative size of the tumor at the 95th percentile of the tumor size distributions versus relative tumor number in the indicated genotypes of mice. The impact of inactivation tumor suppressor genes on tumor number enrichment and tumor size are not correlated. Each dot represents the tumors initiated from one Lenti-sgRNA/Cre vector in the context of the oncogene indicated on the *x*- and *y*- axes. For relative tumor number metrics in all panels in this figure, *Egfr;Cas9* mice are represented by *n* = 18 biologically independent animals, *Braf;Cas9* mice are represented by *n* = 28 biologically independent animals, *G12D;Cas9* mice are represented by *n* = 33 biologically independent animals, and *G12C;Cas9* mice are represented by *n* = 18 biologically independent animals.

**A**

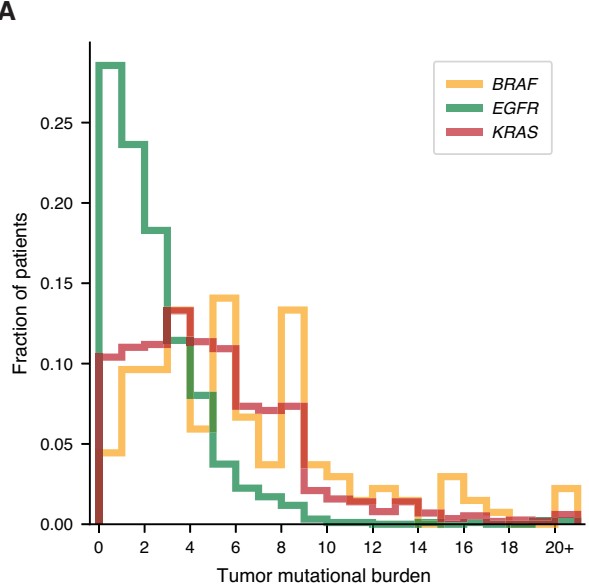

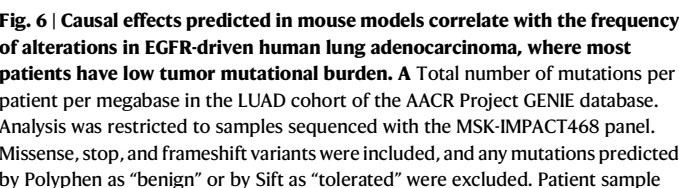

**B**

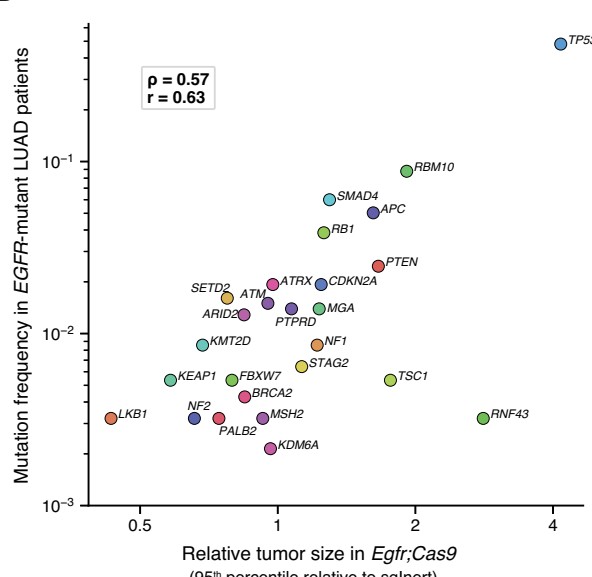

**Fig. 6 | Causal effects predicted in mouse models correlate with the frequency of alterations in EGFR-driven human lung adenocarcinoma, where most patients have low tumor mutational burden. A** Total number of mutations per patient per megabase in the LUAD cohort of the AACR Project GENIE database. Analysis was restricted to samples sequenced with the MSK-IMPACT468 panel. Missense, stop, and frameshift variants were included, and any mutations predicted by Polyphen as "benign" or by Sift as "tolerated" were excluded. Patient sample sizes were: *KRAS n* = 1134, *EGFR n* = 935, and *BRAF n* = 135. The same patients were used for all following human analysis panels. **B** Correlation of relative tumor size at the 95th percentile to co-mutation rate of each gene tested in our model with *EGFR* in LUAD patients. *CMTR2* was the only gene tested in our model that was not present in the MSK-IMPACT468 panel and therefore not included in this analysis. Spearman rank-order correlation (ρ) and Pearson correlation (r) are indicated.

and (ii) the high number of mutations in a tumor can generate a large number of passenger mutations, even in driver genes[20,51].

To minimize these potential confounders, we first aligned the strength of causal effects in our mouse data with the frequency of alterations in human EGFR-driven lung adenocarcinoma. EGFR-driven lung adenocarcinomas have a low tumor mutational burden (TMB)[52] (Fig. 6A), and our mouse data suggest that inactivation of several putative tumor suppressor genes are deleterious and thus unlikely to be observed in the human data, even as passengers. Indeed, there is a strong correlation between mouse cause-and-effect data and mutation frequencies in human EGFR-driven tumors (Fig. 6B; Spearman ρ = 0.57, Pearson r = 0.63). Restricting analysis of human mutation frequencies to patients with EGFR L858R mutations retains the strong correlation between mouse data and human mutational frequency (Spearman ρ = 0.57, Pearson r = 0.67). The genes whose loss is predicted to be detrimental to EGFR-driven tumors (i.e., *Keap1*, *Lkb1*, and *Nf2*) are rarely co-mutated with EGFR in human lung adenocarcinoma as predicted (Fig. 6B). Furthermore, *P53* inactivation, which provides a strong benefit in our mouse models, is very commonly co-mutated with EGFR. Even excluding these extreme examples, the relationship between mouse causal data and human observational data remains strongly correlated (Fig. 6B; Spearman ρ = 0.42, Pearson r = 0.41).

As anticipated, in the KRAS and BRAF contexts where the TMB is generally high there was a poor correlation between co-mutation frequency in human lung adenocarcinoma and causal mouse effects, suggesting that in these subgroups, human mutational frequency does not predict the importance of most tumor suppressor genes (Supplementary Fig. 11A, B). Consistent with previous studies, our analyses showed that in high-TMB tumors the mutation frequency of most tumor suppressor genes is strongly predicted by gene length and thus is very similar between the KRAS and BRAF contexts (Supplementary Fig. 11C, D; Spearman ρ = 0.77 and ρ = 0.82, respectively)[53]. The implication of this observation is that most mutations, even those in functionally important tumor suppressor genes, in KRAS- and BRAF-mutant tumors are in fact passengers. High passenger mutation loads as well

as a variety of mechanisms of tumor suppressor inactivation (beyond direct genomic alteration) together obscure functionally important interactions between oncogene and tumor suppressor alterations that are revealed by in vivo cause-and-effect experiments. For instance, *PTEN* is rarely mutated in KRAS- and BRAF-driven human lung cancers, and yet *Pten* inactivation provides a very strong tumor fitness advantage in our autochthonous mouse models (Fig. 6B and Supplementary Fig. 11A, B). Indeed, the PI3K pathway is commonly activated by non-mutational mechanisms in human lung tumors, and thus *PTEN* and other members of the PI3K/AKT pathway may be important regulators of human lung tumorigenesis[54–57]. This underscores the importance of unbiased functional genomic studies as we have done here, as driver alterations that occur rarely are not necessarily unimportant when present.

## Discussion

In this study, we investigated the fitness landscape of lung tumorigenesis by quantifying the joint effects of inactivating 28 known and putative tumor suppressor genes across four oncogenic contexts on tumor development in vivo (Fig. 7). In total, we quantified the fitness of 112 distinct oncogene by tumor suppressor pairs by assaying the ability of these genetic combinations to initiate tumorigenesis and drive tumor growth. While our previous work defined the fitness landscape of lung tumor suppression in the context of KRAS G12D[19], the landscapes within the three other oncogenic contexts are largely unstudied (Supplementary Fig. 1). Going beyond understanding fitness within isolated oncogenic contexts, our multiplexed and quantitative approach allowed direct comparison of tumor suppressive effects across multiple contexts. Indeed, to our knowledge, we generated data on fifteen times more cross-oncogene tumor suppressor effect comparisons than have been studied previously in quantitative in vivo models (Supplementary Fig. 12). And although we did find alignment between growth effects in our model and human LUAD mutation rates in the context of *EGFR*—an oncogene notable for its low tumor mutational burden—most of the interactions we observed could not have

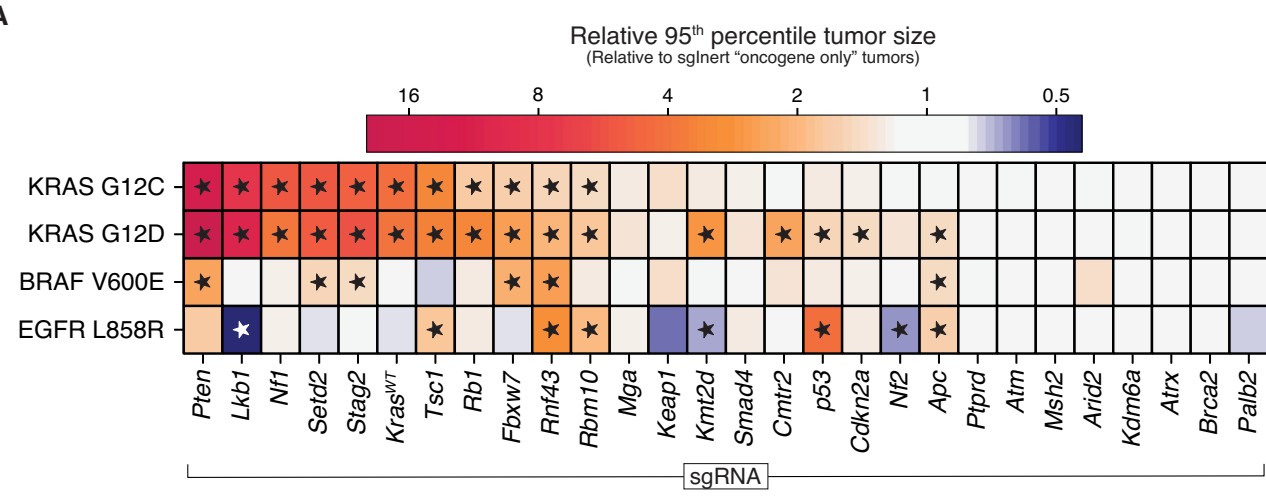

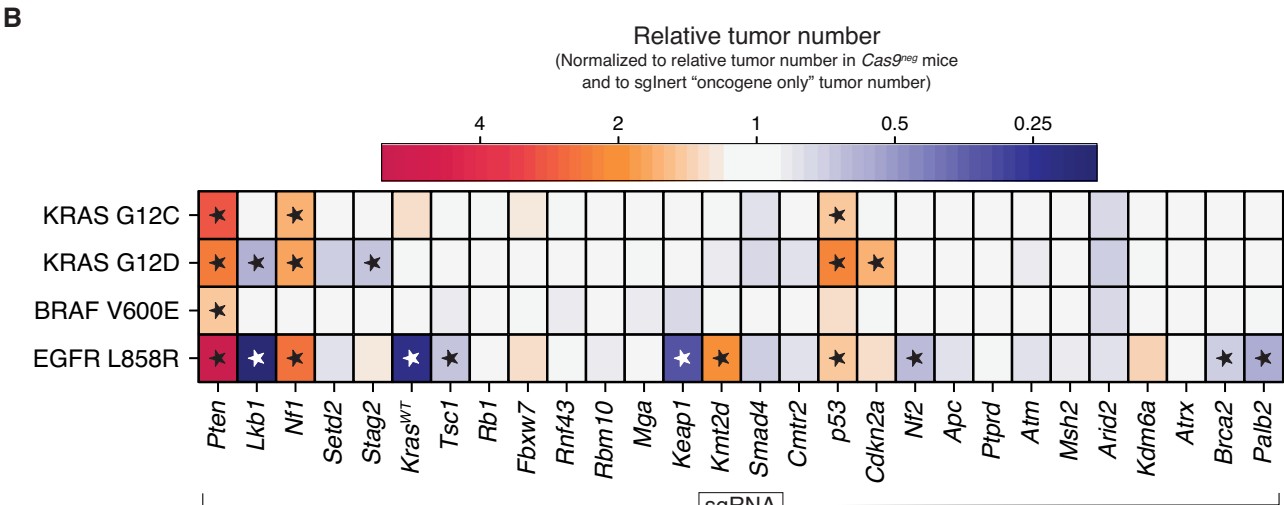

**Fig. 7 | The impact of tumor suppressor pathways on tumorigenesis largely depends on which oncogene is activated and is not predicted by the underlying strength of the oncogene alone. A**, **B** Relative tumor size ratio at the 95th percentile (**A**) and relative tumor number (**B**) for tumors with the indicated Lenti- sgRNA/Cre vector on the x-axis and oncogenic allele on the *y*-axis. Asterisks indicate effects that are significant with FDR at 0.05 and half a log2-fold change from neutral.

been inferred from human data alone, e.g., with mutation rates overwhelmed by passengers for about 80% of the genes studied (Supplementary Fig. 11C, D) in the contexts of KRAS and BRAF.

The scale of our data allowed us to demonstrate that tumor suppressor effects vary strongly by oncogenic context and that the fitness landscape of tumor suppression displays strong and abundant epistasis. Of 28 tumor suppressors studied, few increased size or number across all oncogenes. Only inactivation of one, *Pten*, had a consistently strong, positive effect on both tumor growth and tumor number across all oncogenic contexts. *Rnf43* and *p53* consistently increased relative tumor size and number, respectively (Fig. 7A, B). Given the general trend of widespread epistasis, the robustness of the tumor suppression provided by these three genes is notable. The physiological role of *p53* in constraining tumor initiation/early expansion is striking and might be one reason for the prevalence and ubiquitous nature of *TP53* mutations in human cancer.

Many tumor suppressors showed clear sign epistasis with the oncogenes, whereby inactivation was advantageous in one context and either neutral or deleterious in another context. Surprisingly, inactivation of some of the strongest tumor suppressors in the presence of oncogenic KRAS variants decreased tumor growth in the presence of oncogenic EGFR. Furthermore, the oncogenic contexts were qualitatively different from each other: loss of tumor suppressors generally led to increased rates of tumor initiation and growth in the KRAS backgrounds, had more muted effects in the BRAF context, and had variable effects in the EGFR context.

Some of these epistatic effects were expected given our understanding of the RAS pathway and thus serve as positive controls. For example, the inactivation of NF1—a positive regulator of KRAS GTP to GDP transition—should shift KRAS proteins into their GTP-bound state and increase tumor number and/or growth in the KRAS G12C-, KRAS G12D- and EGFR-driven tumors. However, as class I mutations (e.g., BRAF V600E/D/K/R) have been demonstrated to activate MAPK signaling independent of upstream RAS signaling, inactivation of NF1 was expected to be neutral in BRAF V600E-driven lung tumors[58]. Likewise, inactivation of wild-type KRAS was expected to increase tumor number and/or growth in the KRAS G12C and KRAS G12D contexts, as wild-type KRAS suppresses oncogenic KRAS[33]. Conversely, inactivation of wild-type KRAS was expected to reduce tumor number and/or growth in the EGFR context, as EGFR signals via wild-type KRAS, and, as with NF1-deficiency, should not affect BRAF-driven tumors[22]. Indeed, we observed all of these expected effects in our data, providing an

important validation of our results (Figs. 2A, G, 4E, and 7 and Supplementary Fig. 8D).

The other cases of strong epistasis that we observed could not have been predicted based on the linear oncogenic EGFR→KRAS→BRAF pathway model. It is unclear why inactivation of *Lkb1*, *Setd2*, *Keap1*, *Kmt2d*, or *Nf2* leads to increased growth of oncogenic KRAS-driven lung tumors but is deleterious to EGFR-driven lung tumors (Fig. 7). This pattern indicates that the "off-axis" (i.e., not within the linear RAS pathway) signaling controlled by these three oncogenes drastically shifts the fitness effects of tumor suppressor losses. This should, in turn, affect the set of evolutionary trajectories that are likely after the initial oncogenic events of tumorigenesis.

Furthermore, the fitness effects of subsequent tumor suppressor inactivation could not have been predicted from the basal oncogenic potential of these four oncogenes, from strongest to weakest: KRAS G12D to BRAF V600E to KRAS G12C to EGFR L858R. The precise quantitative similarity of the tumor suppressive effects across KRAS variants is particularly striking because it implies that such effects can be robust to extreme (>tenfold) differences in oncogenic potential, differences in biochemical properties and enzymatic activities[31,34], and diminishing-returns epistasis[7–9]. Overall, it appears that it is not possible to predict the impact of tumor suppressor alterations from simple linear pathway structures or from the fitness effects of the activated oncogenes in isolation.

Due to this lack of predictability, large, multiplexed screens that assay fitness effects across many genetic combinations are critical for revealing the contours of the fitness landscape. But such multiplexing makes it challenging to collect additional data—such as histological staging, immunohistochemistry (e.g., on pERK, cell proliferation, or apoptosis), or spatial or single cell transcriptomics information—which may contribute to our understanding of the biology underlying the observed fitness effects. Future experiments in mice with tumors of a single genotype of interest in which rich readouts from diverse analytes can be measured using existing methods should uncover the mechanistic underpinnings of these genetic interactions.

In addition, this unpredictability may have implications for targeted therapeutic interventions, which work by repressing the signaling of oncogenes or co-linear nodes. If the epistasis also applies in reverse—i.e., that the fitness *costs* of oncogenic signal repression have strong, rugged interactions—then drug effects will be influenced by tumor suppressor inactivation in complex, target-specific ways that will require direct cause-and-effect empirical testing to unravel. Indeed, this has been found in previous studies using Tuba-seq in the context of EGFR inhibitors[36] and chemotherapy[59], provides a possible explanation of why many cancer therapies have low response rates, and suggests that matching patients to therapies based on both oncogene and tumor suppressor alterations may be critical to improving clinical outcomes.

## Methods

### Design and generation of Lenti-sgRNA/Cre vectors

We generated lentiviral vectors encoding Cre and an sgRNA (expressed from a human U6 promoter) targeting each of the following 28 genes, which are known or putative tumor suppressors that are recurrently mutated in lung adenocarcinoma (or pan-carcinoma) and represent diverse cancer pathways:[25,26] *Apc, Arid2, Atm, Atrx, Brca2, Cdkn2a, Cmtr2, Fbxw7, Kdm6a, Keap1, Kmt2d, Kras^{WT}, Lkb1, Mga, Msh2, Nf1, Nf2, Palb2, Pten, Ptprd, Rb1, Rbm10, Rnf43, Setd2, Smad4, Stag2, Tsc1,* and *p53*. Vectors encoding "inert" sgRNAs were also generated: sg*Rosa26-1*, sg*Rosa26-2*, sg*Rosa26-3*, sg*NT-1*, sg*NT-2*, and sg*NT-3* were used in the *G12C;Cas9*, *G12D;Cas9*, and *Braf;Cas9* experiments, while sg*NT-2* and sg*Neo-1* were used in the *Egfr;Cas9* experiments.

sgRNAs were designed and selected as follows. First, all possible 20-bp sgRNAs (using an NGG PAM) targeting each gene of interest were identified and scored for predicted on-target cutting efficiency using an available sgRNA design/scoring algorithm[60]. For each tumor suppressor gene, we then selected the sgRNA predicted to be the most likely to produce null alleles: preference was given to sgRNAs that were previously validated in vivo[20,31,61], had the highest predicted on-target cutting efficiencies, targeted exons conserved in all known splice isoforms (ENSEMBL), targeted splice acceptor/splice donor sites, positioned earliest in the gene coding region, occurring upstream of or within annotated functional domains (InterPro; UniProt), and occurring upstream of or at known recurrent mutation sites in human lung adenocarcinomas. The sgRNA sequences for each target are listed in Table 1.

Twenty-four of these 28 sgRNAs had been validated in our previous work; *Kdm6a*, *Kras^{WT}*, *Msh2*, and *Palb2* are the only new sgRNAs[19,20]. In Cai et al., a rigorous analysis was performed to calculate the expected true positive rate given that sgRNAs targeting the same gene were concordant across multiple metrics and therefore consistent with on-target effects. The true positive rate was above 89% for all metrics tested, and in particular, was above 95% for the relative tumor size metric, which is consistent with the data produced in this study in Supplementary Fig. 5.

To generate Lenti-sgRNA/Cre vectors containing each sgRNA, a double-stranded DNA fragment (IDT gBlock) containing a U6-sgRNA-tracrRNA cassette flanked by restriction sites (AscI and SbfI) was synthesized and digested by AscI and SbfI. This digested DNA fragment was then cloned into an AscI/SbfI-digested parental pLL3.3 lentivector encoding Cre to produce each circularized Lenti-sgRNA/Cre vector.

**Table 1 | Sequence of each sgRNA used in the main experiments**

| sgRNA target | sgRNA sequence |
|---|---|
| Apc | TTGAGCGTAGTTTCACTCCG |
| Arid2 | GGCAGTTCCACCACAGCAGA |
| Atm | GTATCTCAGCAACAGTGGCT |
| Atrx | CAGGTTCATCAAGGTCAAAG |
| Brca2 | GTACCCAAAGTCTCGTCAAG |
| Cdkn2a | CGGTGCAGATTCGAACTGCG |
| Cmtr2 | GTAAGCCACTCGATAATGAG |
| Fbxw7 | ACGTTAGTGGGACATACAGG |
| Kdm6a | TTCCTCATCACCGAAAGCGG |
| Keap1 | TCAAATACGACTGCCCGCAG |
| Kmt2d | TTGTGCTCTCTGTAACTGCG |
| Kras^{WT} | CTTGTGGTGGTTGGAGCTGG |
| Lkb1 | CCACTCTCTGACCTACTCCG |
| Mga | TTATACCGATGACTATCCAC |
| Msh2 | GCGCCGTGTAAAAGTCGCCG |
| Nf1 | CCAAACGTAAAGCAGCAGTG |
| Nf2 | GCTTGGTATGCGGAGCACCG |
| Palb2 | GCACATTGATGACTCCTACC |
| Pten | TCACCTGGATTACAGACCCG |
| Ptprd | CTTGGTGCGGAGCACATCTG |
| Rb1 | TCTTACCAGGATTCCATCCA |
| Rbm10 | GTATTTCCTGAACAGATCCG |
| Rnf43 | TAGACAGATGGCACACACGG |
| Setd2 | TCTCTAATCCATCTTCCCAG |
| Smad4 | GATGTGTCATAGACAAGGTG |
| Stag2 | GGTCAAGAAGCGCTATGTCC |
| Tsc1 | ATCGTGTGGCTCCTGCAAGG |
| p53 | AGGAGCTCCTGACACTCGGA |

**Table 2 | Sequence of each sgRNA used in the validation experiment in Supplementary Fig. 5**

| sgRNA target | 2nd sgRNA sequence |
|---|---|
| *Apc* | CCTTCTACACAGTACACCCG |
| *Arid2* | ACTTGCAGTAAATTAGCTCG |
| *Atm* | GTGAAGTATCTCAGCAACAG |
| *Atrx* | GAATGGCCGTAAAAGTTCTG |
| *Brca2* | AGCTGTTTAAAACACCACAG |
| *Cdkn2a* | GGGCCGCCCACTCCAAGAGA |
| *Cmtr2* | CAGCCTGAATCCATACCACG |
| *Fbxw7* | GTATGTCACAGATTCTAACG |
| *Kdm6a* | ATGGCGGCGGGAAAAGCGAG |
| *Keap1* | CATGTACCAGATTGACAGCG |
| *Kmt2d* | GTTCACCATTAATACCCCCA |
| *Kras*WT | AAACTTGTGGTGGTTGGAGC |
| *Lkb1* | GGGCCTGTACCCATTTGAGG |
| *Mga* | TGACCTCTGATGTACATACG |
| *Msh2* | CCTTAATAAATGCAGCCCGG |
| *Nf1* | GACAAGATGACAAACCTGGT |
| *Nf2* | GACCCCTCTGTGCACAAGCG |
| *Palb2* | ACTGCTGCGCCTAACGACAG |
| *Pten* | TGTGCATATTTATTGCATCG |
| *Ptprd* | AACTCCGGTTGATCAGACAG |
| *Rb1* | AAATGATACGAGGATTATCG |
| *Rbm10* | TGTCGGCCAGGATTCCTACG |
| *Rnf43* | CGTGTGGATCCTCCTGACCG |
| *Setd2* | GCATTCGCTTAATATCCCGG |
| *Smad4* | GGTGGCGTTAGACTCTGCCG |
| *Stag2* | ATTTCGACATACAAGCACCC |
| *Tsc1* | GGAGAGTCAAAGCCCCCTCG |
| *p53* | GAAGTCACAGCACATGACGG |

## Barcode diversification of Lenti-sgRNA/Cre

To enable quantification of the number of cancer cells in individual tumors in parallel using high-throughput sequencing, we diversified the Lenti-sgRNA/Cre vectors with a 46 bp multi-component barcode cassette that would be unique to each tumor by virtue of stable integration of the lentiviral vector into the initial transduced cell. This 46 bp DNA barcode cassette was comprised of a known 6-nucleotide ID specific to the vector backbone (vectorID), a 10-nucleotide ID specific to each individual sgRNA (sgID), and a 30-nucleotide random barcode containing 20 degenerate bases (random BC; Supplementary Fig. 2A).

The 46 bp barcode cassette for each sgRNA was flanked by universal Illumina TruSeq adapter sequences and synthesized as single-stranded DNA oligos. Forward and reverse primers complementary to the universal TruSeq sequences and containing 5' tails with restriction enzyme sites (AscI and NotI) were used in a PCR reaction to generate and amplify double-stranded barcode cassettes for cloning. Each Lenti-sgRNA/Cre vector and its matching insert barcode PCR product was digested with AscI and NotI.

To generate a large number of uniquely barcoded vectors, we ligated 1 μg of linearize vector and 50 ng of insert with T4 DNA ligase in a 100 μl ligation reaction. Four to five hours after incubation at room temperature, ligated DNA was precipitated by centrifugation at 20,000×*g* for 12 min after adding 5 μl Glycogen (5 mg/ml) and 280 μl 100% Ethanol into the ligation reaction. The DNA pellet was washed with 80% Ethanol and air-dried before being resuspended with 10 μl water. This 10 μl well-dissolved DNA was transformed into 100 μl of Sure Electrical Competent Cells using BioRad electroporation system

following manufacturer's instructions. Electroporation-transformed cells were immediately recovered by adding into 5 ml pre-warmed SOC media. From these 5 ml of bacteria, 10 μl were further diluted with LB ampicillin broth, and a final dilution of 1:200,000 was plated on an LB ampicillin agar plate for incubation at 37 °C. The remaining bacteria were mixed gently and thoroughly before being inoculated into 100 ml LB ampicillin broth. After shaking at 37 °C overnight, colony numbers on the LB ampicillin agar plate were counted to estimate the complexity of each library and the 100 ml bacterial culture was pelleted for plasmid purification.

Eight colonies from each library were picked and PCR screened for verification of the specific sgRNA sequence and corresponding barcode sequence among these eight colonies. The final purified library plasmid for each library was again sequence verified.

## Generation of vectors to confirm the correlation of effects of sgRNAs targeting the same gene

To confirm that the effects of sgRNAs targeting the same gene produce correlated effects on in vivo lung tumorigenesis (consistent with sgRNA effects being driven by on-target gene inactivation), we generated an independent set of Lenti-sgRNA/Cre vectors using a pScribe lentiviral backbone (Cellecta; see Supplementary Fig. 5D, E) encoding the same 28 sgRNAs listed above as well as a second unique sgRNA for each of those 28 genes. The unique 2nd sgRNA targeting each gene is listed in Table 2.

## Production, purification, and titering of lentivirus

Twenty-four hours prior to transfection, $2.4 \times 10^7$ 293T cells were plated on a 15-cm tissue culture plate. In total, 30 μg of pPack (packaging plasmid mix) and 15 μg of library plasmid DNA were mixed well in 1.5 ml serum-free D-MEM medium before an equal volume of serum-free D-MEM medium containing 90 μl of LipoD293 was added. The resulting mixture was incubated at room temperature for 10–20 min before adding into 293T cells. At 24 h post-transfection, replace the medium containing complexes with 30 ml of fresh D-MEM medium supplemented with 10% FBS, DNase I (1 unit/ml), $MgCl_2$ (5 mM), and 20 mM HEPES, pH 7.4. The entire virus-containing medium from each plate was collected and filtered through a 0.2 μm PES filter (Nalgene) at 48 h post-transfection. The viruses were further concentrated by centrifugation at 41,325×*g*, 4 °C for 2 h, and the pellet was resuspended in 500 μl PBS buffer. In all, 50 μl virus aliquots were stored at −80 °C.

To quantify the titer of packaged library constructs, $10^5$ LSL-YFP MEF cells[29] were transduced with 1 μl of viruses in 1 ml culture medium containing 5 μg/ml polybrene. Transduced cells were incubated for 72 h before being collected for FACS analysis to measure the percentage of YFP-positive cells. Control viruses were used in parallel to normalize the virus titers.

## Pooling of Lenti-sgRNA/Cre vectors

To generate a pool of barcoded Lenti-sgRNA/Cre vectors for initiation of multiple tumor genotypes within individual mice, barcoded Lenti-sgRNA/Cre vectors targeting the 28 genes described above (*Apc, Arid2, Atm, Atrx, Brca2, Cdkn2a, Cmtr2, Fbxw7, Kdm6a, Keap1, Kmt2d, Kras*WT*, Lkb1, Mga, Msh2, Nf1, Nf2, Palb2, Pten, Ptprd, Rb1, Rbm10, Rnf43, Setd2, Smad4, Stag2, Tsc1,* and *p53*), and those containing the inert, negative control sgRNAs, were combined such that the viruses would be at equal ratios in relation to their estimated in vitro or in vivo titers. The same barcoded Lenti-sgRNA/Cre vectors (same sgRNAs and virus preparations) for each gene were used in all the main experiments. In the *EGFR;Cas9* experiment, some viruses were underrepresented in the pool as we were limited by total volume of those viruses, and in that same experiment, the virus pool contained additional targets for which data were not included in this study. All virus pools were diluted with 1× DPBS to reach the necessary titer for each experiment.

**Table 3 | Details of PCR program**

| Step | Temperature (°C) | Time | Cycles |
|---|---|---|---|
| Initial denaturation | 98 °C | 30 s | |
| Denaturation | 98 °C | **10 s** | ×27 |
| Annealing | 63 °C | **10 s** | |
| Extension | 72 °C | **10 s** | |
| Final extension | 72 °C | 5 min | |
| Hold | 4 °C | ∞ | |

## Mice, tumor initiation, and tissue collection

$Kras^{LSL-G12D}$, $Braf^{CA-V600E}$, $tetO-EGFR^{L858R}$, $Trp53^{flox}$, $Rosa26^{LSL-rtTA3-ires-mKate}$, $Rosa26^{LSL-Cas9-2a-GFP}$ and $H11^{LSL-Cas9}$ alleles have been described[29,35–37,62–64]. Mice with *Kras* and *Braf* alleles were on a BL6 (C57BL/6) background, while mice with the *Egfr* allele were on a mixed BL6/129/FVB background. Mice 6–30 weeks old were anesthetized with isofluorane (1–3% inhaled) and lung tumors were initiated in the mice via intratracheal delivery of a lentivirus pool as previously described[20,65]. Briefly, a catheter was carefully inserted into the trachea of each anesthetized mouse, virus was pipetted into the opening of the catheter, and the catheter was left in place for about one minute while the mouse inhaled the virus. The lentivirus pool contained barcoded Lenti-sgRNA/Cre vectors targeting 28 genes (*Apc, Arid2, Atm, Atrx, Brca2, Cdkn2a, Cmtr2, Fbxw7, Kdm6a, Keap1, Kmt2d, Kras^{WT}, Lkb1, Mga, Msh2, Nf1, Nf2, Palb2, Pten, Ptprd, Rb1, Rbm10, Rnf43, Setd2, Smad4, Stag2, Tsc1,* and *p53*). For all experiments using *G12D;Cas9, G12C;Cas9,* and *Braf;Cas9* mice, the lentivirus pool also contained vectors encoding 6 negative control sgRNAs: three targeting the *Rosa26* gene, which are actively cutting but functionally inert, and 3 non-cutting sgRNAs with no expected genomic target (sgNon-Targeting: sg*NT*). For all experiments using *Egfr;Cas9* mice, the lentivirus pool contained vectors encoding two negative control sgRNAs: one targeting the Neomycin resistance gene within the *Rosa26* allele, which is actively cutting but functionally inert[64], and one non-cutting sgRNA with no expected genomic target (sg*NT*). To induce oncogenic EGFR expression in *Egfr;Cas9* mice, mice were fed doxycycline-impregnated food pellets (625 ppm; HarlanTeklad) starting 1–2 days prior to delivery of pooled barcoded Lenti-sgRNA/Cre vectors. We randomized mice across groups such that sexes and ages were approximately evenly represented in each study group (Supplementary Dataset 1).

Whole lung tissue was extracted from euthanized mice as previously described[20]. Lung mass measurements were recorded as a proxy for overall lung tumor burden. Individual lung lobes from some mice were inflated with 10% neutral buffered formalin and allowed to fix for 16–24 h before passaging into 70% ethanol for subsequent embedding, sectioning, and histological analyses using conventional methods. The remaining lung tissue was weighed and then stored at −80 °C prior to subsequent processing for next-generation sequencing (see sections below).

All animals were kept in pathogen-free housing and animal experiments were conducted in accordance with protocols approved by either the Yale University Institutional Animal Care or Explora BioSciences Institutional Animal Care and Use Committee (IACUC) guidelines. Mice were housed in a pathogen-free environment in Innovive Disposable IVC cages made from 100% high-viscosity PET. Each cage had a dual HEPA-filtered ventilation system. The density of mice was limited to five per cage. Animal rooms had a controlled 12 h light/dark cycle. The normal temperature and relative humidity ranges in the animal rooms were $23 \pm 2.5$ °C and $50 \pm 20\%$, respectively. Cages were set to have 50–60 air exchanges per hour. Water (filtered, purified, and acidified to a pH of 2.5 to 3.0; e.g., Aquavive acidified water from Innovive) and standard rodent chow (e.g., Teklad 2920X irradiated diet) were provided ad libitum.

A veterinarian oversaw and maintained authority over all animal welfare. Mice experiencing pain or distress (or found moribund) as evidenced by prolonged respiratory distress, poor grooming, inability to eat, lack of movement, loss of greater than 10% of their body weight over any window of time, or a rapid or sustained deterioration in health status resulting in a Body Condition Score (BCS)[66] of ≤2 were deemed to require immediate euthanasia. Mice were euthanized using $CO_2$ followed by a secondary method (i.e., cervical dislocation or thoracotomy).

## Generation of spike-in controls

DNA barcode cassettes comprised of 46 bp barcode cassettes and flanked by universal Illumina TruSeq adapter sequences as well as additional buffer sequences to extend their total length to >400 bp were generated either by direct synthesis of the double-stranded DNA fragments (GeneWiz, IDT) or synthesis of single-stranded DNA oligos (GeneWiz, IDT) with overlapping complementary regions that were extended and amplified via PCR to create double-stranded DNA products that were then purified. Aliquots of these stock double-stranded DNA fragments were diluted to the desired copy numbers using DNase-free ultra-pure $H_2O$ and stored at −20 °C.

## Isolation of genomic DNA from mouse lungs

Whole lungs were removed from the freezer and allowed to thaw at room temperature. Spike-ins were added to each whole lung sample. Qiagen Cell Lysis Buffer and proteinase K from Qiagen Gentra PureGene Tissue kit (Cat # 158689) was added as described in the manufacturer protocol. Whole lungs plus spike-ins from each mouse were homogenized in the Cell Lysis buffer and Proteinase K solution using a tissue homogenizer (FastPrep-24 5 G, MP Biomedicals Cat # 116005500). Homogenized tissue was incubated at 55 °C overnight. To remove RNA from each tissue sample, RNase A was added with additional spike-ins to the whole homogenized tissue. To maintain an accurate representation of all tumors, DNA was extracted, and alcohol precipitated from the entire lung lysate using the Qiagen Gentra PureGene kit as described in manufacturer protocol. More spike-ins were added to the resuspended DNA.

## Preparation of barcode libraries for sequencing

Libraries were prepared by amplifying the barcode region from 32 μg of genomic DNA per mouse. The barcode region of the integrated Lenti-sgRNA/Cre vectors was PCR amplified using primer pairs that bound the universal Illumina TruSeq adapters and contained a primer tail encoding unique dual indexes plus Illumina P5/P7 adapters. Specifically, the forward primer was comprised of the P5 Illumina adapter sequence followed by a unique i5 index and then a sequence complimentary to the Illumina TruSeq R1 adapter, as follows (5′ to 3′): AATGATACGGCGACCACCGAGATCTACAC[unique i5 index]ACACTCTTTCCCTACACGA. The reverse primer was comprised of the P7 Illumina adapter sequence followed by a unique i7 index and then a sequence complementary to the Illumina TruSeq R2 adapter, as follows (5′ to 3′): CAAGCAGAAGACGGCATACGAGAT[unique i7 index]GTGACTGGAGTTCAGACG.

We used a single-step PCR amplification of barcode regions, which we found to be a highly reproducible and quantitative method for determining the number of cancer cells in each tumor. We performed eight 100 μl PCR reactions per mouse (4 μg genomic DNA per reaction) using Q5 HF HS 2x mastermix (NEB #M0515) with the following PCR program (Table 3).

The concentration of amplified barcode product in each PCR was determined by TapeStation (Agilent Technologies). Sets of 20-60 PCRs were pooled at equal molar ratios of barcode product, normalized to the estimated burden of tumors (measured lung mass minus an estimated normal lung mass of between 0.15 and 0.18 g) in

each mouse lung sample (measured lung mass minus an estimated normal lung mass of between 0.15 and 0.18 g) associated with the PCRs. Pooled PCRs were cleaned up using a two-sided SPRI bead purification. Samples were sequenced on an Illumina NextSeq 550 or NovaSeq 6000.

## Analysis of sequencing data

Paired-end sequencing reads were demultiplexed via unique dual indexes using BCLConvert (version 3.8.2) and adapters sequences were trimmed using CutAdapt (version 4.1). CutAdapt was used in paired-end mode with the following parameters: minimum-length=0, error-rate=0.1, overlap=3. Paired-end alignments were constructed between mate-paired reads and library-specific databases of the expected oligonucleotide spike-in and tumor barcode insert sequences using Bowtie2 (version 2.4.4, RRID:SCR_016368). These alignments were stringently filtered from downstream analysis if they failed to meet any of several quality criteria, including:

- No mismatches between the two mate pairs, which fully overlap one another, at any location.
- No mismatches between the mate-paired reads and expected constant regions of the barcode or spike-in to which they best align.
- No indels in alignments between mate-paired reads and the barcode or spike-in to which they best align.

Following alignment, errors in paired-end reads were corrected via a simple greedy clustering algorithm:

- Reads were *dereplicated* into read sequence/count tuples, $(s_i, r_i)$
- These tuples were re-ordered from highest to lowest based on their read abundances, $\{r_i\}$.
- This list of tuples was traversed from $i = 1...N$, taking one of the following actions for each tuple $(s_i, r_i)$:

  - If $s_i$ *is not* within a Hamming distance of 1 from any $s_j$ with $j < i$, then $(s_i, r_i)$ initiates a new cluster.
  - If $s_j$ is within a Hamming distance of 1 from some $s_j$ with $j < i$, then it joins the cluster of $s_j$.

The resulting clusters are each considered to represent an error-corrected sequence equal to that of the sequence that founded the cluster with read count equal to the sum of the read counts of the dereplicated reads that are members of the cluster.

A second stage of error correction was performed to remove additional errors. Hamming distance $D(s_i, s_j)$ was computed on all pairs of error-corrected sequences. Then, each sequence $s_i$ (with $r_i$ reads) was absorbed into the most abundant sequence $s_j$ (with $r_j > r_i$ reads) if either of the following criteria were met:

- $D(s_i, s_j) \leq 3$
- $D(s_i, s_j) \leq 5$ and $r_j/r_i \geq 5$ or $r_i \leq 3$

These heuristics were established based on internal control data. After applying both rounds of error correction, we estimate a false positive rate of $1.4 \times 10^{-8}$ based on the number of reads assigned to spike-in oligonucleotide sequences (which have no degenerate bases) that were not added to the samples. Following error correction, a filter was applied to remove sequences that could have originated from cross-contamination: barcodes were compared across samples in the same study, and any exact sequences that were found in more than one library were removed.

Following error correction and cross-contamination removal, the read counts of each unique barcode were converted to neoplastic cell number by dividing the number of reads of the spike-in oligonucleotide added to the sample prior to tissue homogenization and lysis at a fixed, known concentration.

## Removal of mice that did not get sufficient viral titer during transduction

Following the sequence processing, mice were removed if they did not reach a lower bound of total neoplastic cells. For the experiments with *G12D;Cas9* and *G12C;Cas9*, mice were removed if they had less than $10^6$ total neoplastic cells. For the experiments with *Braf;Cas9* and *Egfr;Cas9* mice, mice were removed if they had less than $10^5$ total neoplastic cells. Thresholds were chosen using by examining the distribution of total neoplastic cells per mouse across each study. Most mice fall within ~two orders of magnitude of each other, and any outliers fell at least an order of magnitude below the rest of the distribution.

## Accounting for processed lung mass when normalizing metrics by titer

Because several mice had lobes taken for histology and therefore only a fraction of the lung made it into Tuba-seq, the processed lung should not be expected to represent the full viral titer transduced to the mouse. To correct the titer for that fraction of lung, we multiplied the total titer given to the mouse by the ratio of the processed lung weight to the total lung weight before any lobes were removed. This effective titer was used for all plots that present titer-normalized quantities (e.g., Figs. 1F–I and 3D–G).

## Calculation of tumor size percentiles

First, tumors were pooled across all mice in the group, and separated into tumors that map to each Lenti-sgRNA/Cre guide. Tumors from Lenti-sgInert/Cre were pooled (sg*NT*-1, sg*NT*-2, sg*NT*-3, sg*R26*-1, sg*R26*-2, sg*R26*-3 for experiments using *G12D;Cas9*, *G12C;Cas9*, *Braf;-Cas9* mice, and sg*NT*-2, sg*Neo*-1 for the experiments using *Egfr;Cas9* mice) to create one pool of sgInert tumors. Using the sgInert tumors, a minimum tumor size cutoff was determined, above which tumor percentiles would be calculated. The goal of matching this cutoff across study groups, particularly when comparing across oncogenes, was to ensure that the tumor suppressor effects were being measured on the same fraction of initiated tumors, independent of the strength of the oncogene. A cutoff was chosen for each study group that matched the number of sgInert tumors per titer above the cutoff. The exception was for the *Braf;Cas9* experiments. Because the *Braf;Cas9* tumor sizes differed so strikingly from those in the *G12D;Cas9*, *G12C;Cas9*, *Egfr;Cas9* experiments, suggesting a very different process for tumor initiation and/or growth, we opted to use an ad hoc cutoff that captured >85% of total tumor burden and reduced the high mouse-to-mouse variability in the number of small tumors. For the main experiments where all mice of each oncogene-timepoint pair were pooled, the following minimum cutoffs were used: 1600 cells for *G12D;Cas9* 15 weeks, 600 cells for *G12D;Cas9* at 9 weeks, 400 cells for *G12C;Cas9* at 15 weeks, 300 cells for *G12C;Cas9* at 9 weeks, 300 cells for *Egfr;Cas9* at 15 weeks, and 3000 cells for *Braf;Cas9* at 15 weeks. Neoplastic cell number cutoffs for the comparisons of the replicate study groups in Fig. 2 were calculated using the same procedure.

For each set of Lenti-sgRNA/Cre tumors in each oncogene-background pair, size percentiles of tumors above the cutoff were computed and divided by the same size percentiles for the sgInert tumors in the same context with the same cutoff. This ratio is referred to as relative tumor size in Figs. 2, 4, and 7, and Supplementary Figs. 4, 6, and 8. Mice and tumors were bootstrapped 8000 times and the calculation was repeated each time. A 95% confidence interval from these bootstraps was reported.

## Comparison of relative tumor size between Cas9-negative vs Cas9-positive

To confirm that the presence of Cas9 did not affect tumor size of sgInert tumors, we calculated the relative tumor size of non-targeting and active-cutting sgInert tumors across four

independent studies of *G12D;Cas9* mice (Cas9-positive) that each had a group of *G12D* mice (Cas9-negative) (Supplementary Fig. 4D). A minimum tumor size cutoff of 300 was used for all groups, and the metric was calculated as described in Methods Section "Calculation of tumor size percentiles", where the inert tumors in the Cas9-positive group were used for the numerator of the ratio and the Cas9-negative were used for the denominator. In three of the four groups, the relative tumor sizes of the sgInert tumors were the same in Cas9-positive and Cas9-negative mice, with no significant difference at any percentile of the distribution. In the fourth study group, the inert tumors in the Cas9-positive mice were slightly larger than in the corresponding Cas9-negative group, but the magnitude of this effect was small and considered neutral by the metrics used for the rest of the tumor suppressor inactivations. In addition, because the metrics used to measure growth effects are calculated between tumors that are within the same Cas9 background, any effects of Cas9 on tumor size are controlled.

## Calculation of tumor number enrichment

Tumor number enrichment estimates the factor by which there are more or fewer sgRNA tumors above a minimum size cutoff than there would have been if the tumors had been sgInert. As we sought to measure tumor number effects associated with initiation and very early tumor growth, a cutoff of 500 cells was used, as it represents a lower bound on our technical resolution.

First, all tumors from each mouse in each study group being compared are pooled and separated into tumors that map to each Lenti-sgRNA/Cre guide. Tumors from Lenti-sgInert/Cre were pooled (sg*NT*-1, sg*NT*−2, sg*NT*−3, sg*R26*−1, sg*R26*−2, sg*R26*−3 for experiments using *G12D;Cas9*, *G12C;Cas9*, *Braf;Cas9* mice, and sg*NT*−2, sg*Neo*−1 for the experiments using *Egfr;Cas9* mice) to create one pool of sgInert tumors.

Because tumor number for a given sgRNA will be proportional to the titer of the individual sgRNAs within the viral pool, we calculated the ratio of sgRNA tumors to sgInert tumors in mice without Cas9 in each virus pool, which is expected to be driven only by titer differences between viruses.

Then, to calculate the tumor number enrichment for each sgRNA, we divided the number of sgRNA to sgInert tumors, and divided this ratio by the same ratio in the mice without Cas9 that had been initiated with the same pool of viruses. This ratio of ratios is referred to as *relative tumor number* in Figs. 5 and 7 and Supplementary Figs. 9 and 10. Mice and tumors were bootstrapped 8000 times and the calculation was repeated each time. A 95% confidence interval from these bootstraps was reported.

## Calculation of tumor burden densities

The density of tumor burden as a function of log tumor size was estimated as follows. First, we pooled tumors across all mice in each cohort and computed total tumor burden by summing the sizes of all tumors. We then generated log-spaced bins with ten bins per order of magnitude of tumor size, summed the sizes of all Lenti-sgInert/Cre tumors in each bin, and divided by total tumor burden. To create a density in log size, we then divided this ratio by log (bin width), which was a constant given the log-spaced binning. Finally, mice and tumors were bootstrapped 1000 times and this procedure repeated each time, and the mean density across bootstraps as well as a 95% confidence interval are shown for each size.

## Calculating lengths of gene coding regions

For each gene of interest, we used the coding sequence annotations of the "Ensembl Canonical" transcript (Ensembl project, release 105)[67] to determine the length of the gene's coding region.

## Determining gene mutation and co-mutation rates from human lung cancer genomics data

Mutation rates in human LUAD were estimated using AACR Project Genie (release version Genie 12.0)[26]. First, we restricted our analysis to patients with LUAD and then selected those with the relevant oncogene mutations. We used the following definitions of *KRAS*, *EGFR*, and *BRAF* oncogene mutations:

- *KRAS*: any mutation in codon 12, 13, or 61
- *EGFR*: p.L858R, p.L861Q, p.G719X, deletion or insertion in exon 19, insertion in exon 20
- *BRAF*: all mutations listed in Table 1 of Owsley et al.[68].

To minimize bias due to the variety of genetic panels used in AACR Project Genie, we restricted our analysis to patients sequenced with the MSK-IMPACT468 panel, which was the most commonly used panel for LUAD patients. This resulted in 1134 patients with *KRAS* mutations, 935 with *EGFR* mutations, and 135 with *BRAF* mutations. *CMTR2* was the only gene that we tested that was not included in the panel, so we excluded it from our analysis. *KRAS*^WT was also excluded from this analysis because of the difficulty in distinguishing mutations in a wild-type allele from mutations in the oncogenic allele in the Genie data. Correlations of mutation frequencies and mouse effects were also assessed using all panels in the AACR Project Genie database, and Spearman and Pearson correlations produced were similar to those using only MSK-IMPACT468.

To determine the co-mutation frequencies for each of the tumor suppressor genes we inactivated in our mouse models, we counted mutations as follows: first, we selected all mutations that were non-sense, missense, or frameshift variants. Then, any mutations that were predicted by Polyphen to be benign or predicted by SIFT (RRID:SCR_012813) to be tolerated were excluded. Correlations between causal mouse effects and human co-mutation frequencies were also tested with other definitions, and we confirmed that as we move from the set of mutations defined above to non-synonymous mutations, and then to all mutations the Spearman correlations decreased slightly, but the trend remained the same.

Tumor mutational burden was calculated as the total number of mutations per 1,000,000 base pairs. The total gene length was calculated using the sum of exon length for all genes that were queried in the panel.

When correlating exon length and co-mutation frequency, outliers were first removed. We found the same set of outliers using two methods: (1) Clustering was run using the pam algorithm, a robust version of Kmeans, and the gap statistic was calculated using the "globalSEmax" method, which looks for the first value that is lower than the global maximum minus the standard error when evaluated at that value. (2) Spearman correlation was calculated using the full set of genes, and then genes were removed one at a time to test how the correlation increased. Each iteration, the gene whose removal maximized the Spearman correlation was removed. A plot of the number of genes removed against Spearman correlation showed an elbow at the removal of 5 genes. Both methods determined *CDKN2A*, *TP53*, *KEAP1*, *LKB1*, and *RBM10* to be the outliers. Linear regression on a log-log scale was then performed on the remaining genes. Pearson correlation and Spearman correlation were reported on the plot.

Relative tumor size at the 95th percentile was used as the metric of mouse causal effects. Correlation of co-mutation frequency in *EGFR*-driven tumors with relative tumor size at 90th, 75th, and 50th percentiles maintained the trend but became less strong as the percentiles increased, with Spearman correlations of 0.53, 0.49, and 0.43, respectively.

## Statistical analysis

All statistical tests were done in Python, using Scipy version 1.8.1.

## Study design statistics and reproducibility

No sample size calculation was performed for this study. However, prior studies with comparable sample sizes revealed many significant effects[19,20,61]. Further, our bootstrap confidence intervals capture uncertainty in statistical metrics due to random sampling of mice and tumors.

Mice were randomized into experimental groups within each oncogenic background, and sex ratios were all approximately balanced.

Blinding was not relevant to our study because all mice received the same treatment, and no drugs were administered.

As described above, mice were excluded if they did not receive sufficient viral titer during transduction, as measured by tumor barcode sequencing. Although the absolute cutoff was not pre-determined, the method by which we would remove the mice was determined in advance of the study.

Many experiments were done to reproduce our results. We reproduced the tumor suppressor effects in the *G12D;Cas9* mice with 11 different study groups (243 mice) and the tumor suppressor effects in the *G12C;Cas9* mice with 4 different study groups (47 mice).

## Reporting summary

Further information on research design is available in the Nature Portfolio Reporting Summary linked to this article.

## Data availability

The raw tumor data have been deposited in Dryad[69]. The data plotted in the main manuscript, including summary statistics and their confidence intervals, are provided in the Source Data file. Source data are provided with this paper.

## Code availability

The core software used is described in the Methods, under the subsection Analysis of sequencing data. The open-source software includes the following external tools: BCLConvert v3.8.2, CutAdapt v4.1 and Bowtie2 v2.4.4. A description of all open-source code is included in "Methods", and further details are available on request. The proprietary portions of the code are not available. Statistics and plots were generated using Python. Statistical tests were run using Scipy, v1.8.1.

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

## Acknowledgements

We are grateful to all members of D2G Oncology for expert advice and helpful comments. We thank Explora and The Jackson Laboratory for expert animal care. We would like to acknowledge the American Association for Cancer Research and its financial and material support in the development of the AACR Project GENIE registry, as well as members of the consortium for their commitment to data sharing (interpretations are the responsibility of the study authors). We thank AstraZeneca, Bristol-Myers Squibb, Revolution Medicines, Merck KGaA, and CureTeq for generously allowing the inclusion of data generated in collaboration with D2G Oncology. L.E.D. is the Burt Gwirtzman Research Scholar in Lung Cancer. D.A.P. and M.M.W. are supported by NIH R01-CA 234349. This work was supported in part by NIH SBIR R44-CA250672.

## Author contributions

J.M.J., M.M.W., K.P., M.J.R., and I.P.W. conceptualized and designed the study. L.M.B., J.M.J., L.S., V.B.T., W.N., G.D.W., I.K.L., G.F., M.D.C., M.M.W., M.J.R., and I.P.W. curated data. L.M.B., D.A., and M.J.R. conducted formal analyses. K.P., J.J.M., D.A.P., M.M.W., and

I.P.W. acquired funding. L.M.B., J.M.J., L.S., V.B.T., W.N., I.K.L., E.A.A., G.G., D.A., G.F., M.D.C., Z.U., K.P., M.M.W., M.J.R., and I.P.W. conducted research and investigation. L.M.B., J.M.J., G.F., D.D., A.C., M.P.Z., L.E.D., K.P., D.A.P., M.M.W., M.J.R., and I.P.W. designed and developed methodology. L.M.B., J.M.J., I.K.L., G.F., K.P., D.A.P., M.M.W., M.J.R., and I.P.W. conducted project administration. D.D., A.C., M.P.Z., and L.E.D. provided resources. L.M.B., G.D.W., M.G., and M.J.R. designed and developed software. J.M.J., G.D.W., K.P., J.J.M., D.A.P., M.M.W., M.J.R., and I.P.W. provided supervision. L.M.B., G.D.W., M.J.R., and I.P.W. validated results. L.M.B., M.M.W., M.J.R., and I.P.W. designed and created visualizations. L.M.B., D.A.P., M.M.W., M.J.R., and I.P.W. wrote the original draft, and all authors contributed to review and editing of the manuscript.

## Competing interests

L.M.B., J.M.J., L.S., V.B.T., G.D.W., M.G., I.K.L., E.A.A., G.G., D.A., J.J.M., and M.J.R. are current or former employees and shareholders of D2G Oncology. I.P.W. is a co-founder, employee, and shareholder of D2G Oncology. D.A.P. and M.M.W. are co-founders, shareholders, members of the board of directors, and compensated scientific advisors of D2G Oncology. I.P.W., D.A.P., and M.M.W. are co-inventors of patents relating to technologies for autochthonous mouse models of human cancer, which D2G Oncology has exclusively licensed from Stanford University. D.D. and A.C. are employees and shareholders of Cellecta. L.E.D. is a scientific advisor and holds equity in Mirimus. L.E.D., M.P.Z., and Cornell University have licensed the technology described in this manuscript. K.P. is co-inventor on a patent related to EGFR T790M mutation testing issued, licensed, and with royalties paid from MSKCC/MolecularMD. K.P. reports grants to her institution from Boehringer Ingelheim, AstraZeneca, Roche/Genentech, and D2G Oncology, and consulting fees from AstraZeneca and Janssen. The remaining authors declare no competing financial interests.
