## [Peer Review File · Nature Communications]

REVIEWER COMMENTS

Reviewer #1 (Remarks to the Author): expertise in Crispr screening

Blair and colleagues report the application of the Tuba-Seq method to characterize the tumor suppressor landscape of four mouse model of lung adenocarcinomas. The oncogenic drivers for these models were two distinct K-Ras conditional knock-in alleles (G12D and G12C), a dox-inducible EgfrL858R allele, and a conditional BRAf-V600E allele. Despite being considered components of the same signal transduction cascade, (EGFR-RAS-BRAF-MAPK), the authors convincingly show that these three oncogenes display significantly different tumor-suppressive landscapes, in some instances responding to inactivation of the same tumor suppressor in opposite manners. For example, loss of Keap1is slightly promotes tumorigenesis in the BRAf model, while substantially impairing the ability of EgfrL858R to drive lung adenocarcinoma formation. Conversely, while loss of p53 potentially cooperates with EgfrL858R, it has a very modest effect on Braf-V600E.

Even more strikingly, inactivation of Lkb1, a potent tumor suppressor in the KRas models, profoundly reduced tumor formation in the Egfr model.

Overall, the data presented in this manuscript emphasize the complexity of oncogenic pathways and will greatly help future mechanistic studies to dissect them. As such, they will be of great interest to the broad readership of Nature Communications.

The manuscript is extremely well written and was a pleasure to read. The relevant literature is cited, the figures are clear and beautifully presented, the experimental design is elegant and includes the appropriate controls. The statistical methods applied, are, as far as I can judge, appropriate.

I have no major or minor criticisms or suggestions for the authors and I recommend publication of this work in Nature Comm.

Reviewer #2 (Remarks to the Author): expertise in Crispr application within lung cancer

Comments

The work presents a rigorous mouse genetic study that aims to systematically quantify the pair-wise functional interaction between four major oncogenes (KRAS_G12D, KRAS_G12C, BRAF_V600E, and EGFR_L858R) with 28 tumor suppressor genes (TSGs) that are commonly mutated in human lung adenocarcinomas (LUAD). From human data, it is known that some oncogene-TSG interactions are context-specific; for example, LKB1 and KEAP1 mutations tend to co-occur with KRAS mutations, whereas they tend to be mutually exclusive with EGFR mutations. In the past, such functional interactions have been studied in GEMM of lung tumors one gene at a time. This study utilized the power of in vivo CRISPR KO library screen to systematically interrogate over 100 pair-wise interactions to establish a fitness map of these commonly observed genetic lesions in LUAD. Some genetic interactions, particularly in the context of BRAF and EGFR oncogene, have not been formally tested before with traditional GEMMs. Functional data from this work can guide the future analysis of oncogene-TSG interactions. Although the authors carried out extensive genetic screens, there was little validation or mechanism data. The authors' main conclusion is that TSGs cooperate with oncogenes in a context-specific manner. This is generally appreciated already, and the authors demonstrated this principle here in a more comprehensive manner here. Thus, the main weakness of the study, in its current form, is a lack of validation data to verify the genetic interactions observed in the primary screen, and a lack of mechanistic insight on these genetic interactions. It is somewhat unclear what new knowledge has this genetic screen taught us about LUAD, and how this might deepen our understanding of the risk and treatment of human lung cancer.

Major points

1. The authors constructed 2 small libraries of sgRNAs targeting 28 TSGs with 1 guide per gene. It is somewhat surprising that the authors didn't attempt to validate the KO efficiencies of these sgRNAs in vitro, for example, in mouse fibroblasts or lung cancer cell lines at low MOI of < 1.

Knowing all sgRNAs are effective at target knockdown will greatly inform the interpretation of neutral genetic interactions found in the tumor study. As it stands, it is unclear that when a TSG had no impact on tumor growth, whether this is simply because the sgRNA failed to KO the target gene efficiently in cells. Without sgRNA KO validation, the negative data in this study is difficult to interpret, and the authors cannot be confident at concluding many of the TSGs had no appreciable impact on tumor growth.

2. The analysis presented in Figure 5 and Supp Figures 9 & 10 is confusing and doesn't make great sense to me. If I understood correctly, the authors tried to "normalize" the impact of a TSG sgRNA on tumor growth by looking at its effect in Cas9NEG mice, and then deriving a ratio of its relative effect between Cas9+ and Cas9NEG mice. One would expect that none of the sgRNAs to have any effect in Cas9NEG mice. So, any impact on tumor growth in the Cas9NEG mice must be attributable to non-specific effect of a sgRNA through an CRISPR-independent mechanism. It was further confusing that the authors also switched to a different criterion for analyzing tumor fitness. In earlier analyses, the authors used a 50%, 75%, 90% and 95% tumor size to show the effect of a TSG sgRNA relative to control sgRNAs. Here the authors decided to use a relatively arbitrary tumor size cut-off to evaluate the effect of a TSG sgRNA. Why not use the same analysis as before? The data shown in Supp Figure 9A illustrates this problem. In the Cas9NEG mice, sgPten appears to strongly reduce tumor growth (0.3), whereas in the Cas9+ mice sgPten had almost no effect on tumor growth (1.1). This is in stark contrast to the data shown in Supp Figure 2C. It is unclear whether the sgPten data in these two figure panels represent the same experiment or independent repeats, or it results from a change in the data analysis method. Biologically, it might have made more sense to test the oncogene-independent effect of TSG sgRNAs in Cas9+ mice without Kras etc. Although this may result in tumor incidence too low to be informative in a library setting. Thus, it is not clear to me if the analysis in Figure 5 makes biological and technical sense, and I suggest this be removed from the paper.

3. A major weakness of the study in its current form is the lack of validation experiments beyond the in vivo library screen. The authors showed that the G12D and G12C alleles of Kras have different tumorigenic potential in this model, and the TSGs Kmt2d or Cmtr2 cooperate with G12D but not G12C to further enhance tumor growth (Figure 2). This finding, however, was not followed up by validation experiments, for example, using individual sgRNAs to examine lung tumor size and tumor proliferation/apoptosis markers. Mechanistically, how do these two genes cooperate with G12D but not G12C? In addition, in human LUAD data, do mutations these two TSGs co-occur with KRAS G12D but not KRAS G12C? Similarly, although the authors have uncovered new functional interactions with BRAF and EGFR, for example, RNF43, there are no follow-up validation experiments to further support these findings. In human LUAD, RNF43 mutation tends to co-occur with BRAF mutation, yet it tends to be mutually exclusive with EGFR mutation. Thus, validation experiments using single sgRNA would be very helpful to understand the function of RNF43 and other novel TSG-oncogene interactions.

4. In this study, PTEN was identified as the strongest cooperating TSCs. The authors suggested that, together with prior studies, this result supports a major role of the PI3K pathway in human LUAD. I don't feel this argument is sound. None of the PI3K pathway genes, including PTEN and PIK3CA, are frequently mutated in human LUAD, neither have PI3K inhibitors demonstrated efficacy in LUAD in clinical studies. It is more likely that the strong synergistic effect between sgPten and Kras is an "artifact" of mouse models and reflects the fundamental difference between LUAD development in GEMM and in human. We should acknowledge such limitation of mouse models, rather than forcing the issue and use mouse models to implicate genetic interactions that are not necessarily significant in human tumors.

Minor points

1. The authors had constructed 2 libraries with distinct sgRNAs for each TSG. It was unclear which library was used in which experiment. Since neither library appeared to be validated for sgRNA KO efficiency, it might be hard to directly compare experimental data obtained between these two libraries.

2. Does Cas9 alter tumor burden? For example, if the tumor size for inert sgRNAs are compared between Kras G12D and Kras G12D Cas9 mice (Figure 9A), is there a significant difference?

3. Figure 6B, it was unclear what the co-mutation rate (y-axis) measures. The authors seem to suggest that the stronger the genetic interaction, the more likely the TSG is to co-occur with EGFR mutation. If so, it would make more sense to plot the log₂ odds ratio and q-value for co-occurrence.

4. Line 351, "(Fig. 6; Supplementary Fig. 11A and B)" reference is a typo?

Reviewer #3 (Remarks to the Author): expert in lung cancer mouse models

Blair and colleagues assessed the lung tumorigenic effects of CRISPR/Cas9-mediated inactivation of 28 putative tumor suppressor genes in the context of 4 oncogenic contexts: KRAS G12D, KRAS G12C, BRAF V600E, and EGFR L858R. To date, this study likely represents the largest, most comprehensive in vivo analysis of cross-oncogene tumor suppressor effect comparisons. The authors conclude that the fitness landscape is rugged (i.e. the effect of tumor suppressor inactivation often switches between beneficial and deleterious depending on the oncogenic context) and shows no evidence of diminishing-returns epistasis within variants of the same oncogene. The findings presented here suggest that a simple linear oncogenic signaling relationship of EGFR to KRAS to BRAF does not exist because off-axis signaling likely contributes to determining the fitness effects of inactivating tumor suppressors.

This manuscript is well-written and represents a complete narrative that has been strengthened by comprehensive, well-controlled experiments. The data is presented nicely, and the scientific rigor is robust.

1. In Figure 5C, what was the rationale for analyzing Braf;Cas9 relative to Cas9negative (G12D) as opposed to Braf;Cas9 relative to Cas9negative (Braf) as done in panels 5A, 5B, and 5D for G12C, G12D, and Egfr, respectively?

2. Methods: Please include much greater details regarding the intratracheal delivery of the lentiviral to the mice, including anesthesia methods, lentivirus dose, numbers of mice, age of mice, etc. The current description is not comprehensive enough that other researchers could replicate it based on the methodology outlined.

3. The Discussion section currently lacks depth and requires greater assessment of the findings presented here within the context of existing knowledge. How can the conclusions of this manuscript translate to improved therapies for lung cancer patients? What opportunities for future investigation do these results create? What are the limitations of this current study and how could they be overcome?

Reviewer #4 (Remarks to the Author): expert in lung cancer genomics and evolution

The study by Blair and colleagues used various autochthonous mouse models of lung adenocarcinoma (LUAD) driven by KrasG12D, KrasG12C, BrafV600E and EgfrL858R, barcoded lenti-sgRNA/Cre vectors targeting in each of the four models 28 known and putative tumor suppressors, and the tuba-seq previously developed by the group and which quantifies independent tumor clones by deep sequencing of barcodes present in each genomically-integrated lentivirus. The study found that KrasG12D leads to larger and faster lung tumor development, irrespective of the inactivation of tumor suppressors. Inactivation of various tumor suppressors had, for the most part, similar effects on KrasG12D and KrasG12C tumors -- although they did find some disparate functional effects for some, Kmt2d loss enhanced KrasG12D pathogenesis. Braf and Egfr mutant tumors had distinct effects on tumor development - Braf mutant mice had higher burden than those with mutant Egfr and their size were more or less uniform. The authors conclude that the four drivers have distinct tumor developmental potential with KrasG12D displaying the highest potential. Interestingly, loss of some tumor suppressors (e.g. Lkb1) showed opposing effects between Kras and Egfr mutant tumors - Lkb1 loss increased tumor size in KrasG12D models but decreased tumor size in Egfr mutant mice. Similar analysis was performed when comparing to Cas9 negative cells to determine effects of tumor suppressor loss on early stages of tumor development. Interestingly, at least in Egfr mutant tumors, tumor sizes by tumor suppressor loss correlated with co-mutation rate of the tumor suppressor (e.g., Lkb1, Keap1) with EGFR in human LUAD patients. On the other hand, Pten loss yields similar effects on tumor development across different oncogene-driven models. Lastly, the study summarizes effect of

tumor suppressor loss on tumor size and number in the context of activation of the four oncogenic drivers.

The study will pique interest in the field of LUAD development since it highlights oncogene-tumor suppressor interactions that would not be readily observed/interrogated in human LUAD. For instance, Pten is very rarely co-mutated with Kras in human tumors, and the study clearly demonstrates tumor promoting effects of Pten loss in mouse models with mutant Kras activation. There are some comments that need to be addressed prior to publication for the astute readership of Nature communications.

-It is not clear why the study did not first use animals with control lenti-sgRNA to compare and contrast tumor development between KrasG12D and KrasG12C mice.

-On that theme, there appears to be no difference in tumor burden when assessing KrasG12C/D mice with control lent-sgRNA or sgRNA targeting the 28 tumor suppressors - perhaps suggesting that few tumor suppressors significantly impact tumor initiation by oncogenic Kras. Indeed this appears to be the case upon inspection of individual tumor suppressor sgRNAs (e.g., Figure 2). The authors need to discuss this point.

-The authors conclude that Braf mutation, relative to mutant Egfr, leads to more tumor numbers. There is inconsistency between panels C and say E in figure 3. The histopathological images in panel C suggest high tumor number and burden in Egfr;Cas9 mice.

-While the use of the tuba-seq platform is commendable and useful, there are lost opportunities to determine how loss of tumor suppressors not only impacts tumor number and size but also the histological grade/stage of the lesion (hyperplasia versus adenoma versus adenocarcinoma). It will be important to determine by comprehensive histopathological analysis how tumor suppressor loss impacts pathologic progression of lung lesions in the context of the four oncogenic drivers.

-On that theme it is important to validate in resultant (if available and not entirely sequenced, say in additional mice) key markers (e.g., phosphorylated ERK in case of Kras signaling) that may inform of the mechanisms underlying effects of tumor suppressor loss on oncogene-driven lung tumor development.

RESPONSE TO REVIEWERS' COMMENTS

Reviewer #1

Blair and colleagues report the application of the Tuba-Seq method to characterize the tumor suppressor landscape of four mouse model of lung adenocarcinomas. The oncogenic drivers for these models were two distinct K-Ras conditional knock-in alleles (G12D and G12C), a dox-inducible EgfrL858R allele, and a conditional BRAf-V600E allele. Despite being considered components of the same signal transduction cascade, (EGFR-RAS-BRAF-MAPK), the authors convincingly show that these three oncogenes display significantly different tumor-suppressive landscapes, in some instances responding to inactivation of the same tumor suppressor in opposite manners. For example, loss of Keap1 is slightly promotes tumorigenesis in the BRAf model, while substantially impairing the ability of EgfrL858R to drive lung adenocarcinoma formation. Conversely, while loss of p53 potentially cooperates with EgfrL858R, it has a very modest effect on Braf-V600E.

Even more strikingly, inactivation of Lkb1, a potent tumor suppressor in the Kras models, profoundly reduced tumor formation in the Egfr model.

Overall, the data presented in this manuscript emphasize the complexity of oncogenic pathways and will greatly help future mechanistic studies to dissect them. As such, they will be of great interest to the broad readership of Nature Communications.

The manuscript is extremely well written and was a pleasure to read. The relevant literature is cited, the figures are clear and beautifully presented, the experimental design is elegant and includes the appropriate controls. The statistical methods applied, are, as far as I can judge, appropriate.

I have no major or minor criticisms or suggestions for the authors and I recommend publication of this work in Nature Comm.

We thank this Reviewer for their astute summary and complimentary review of our manuscript. This Reviewer commented that the “experimental design is elegant and includes the appropriate controls” and "the data presented in this manuscript emphasize the complexity of oncogenic pathways and will greatly help future mechanistic studies to dissect them.” We share in the Reviewer’s excitement that the features of the oncogene-tumor suppressor landscape described in this manuscript provoke, and serve as a foundation for, follow-up studies to uncover the mechanistic underpinnings of these complex and diverse gene-gene interactions in cancers.

The work presents a rigorous mouse genetic study that aims to systematically quantify the pair-wise functional interaction between four major oncogenes (KRAS_G12D, KRAS_G12C, BRAF_V600E, and EGFR_L858R) with 28 tumor suppressor genes (TSGs) that are commonly mutated in human lung adenocarcinomas (LUAD). From human data, it is known that some oncogene-TSG interactions are context-specific; for example, LKB1 and KEAP1 mutations tend to co-occur with KRAS mutations, whereas they tend to be mutually exclusive with EGFR mutations. In the past, such functional interactions have been studied in GEMM of lung tumors one gene at a time. This study utilized the power of in vivo CRISPR KO library screen to systematically interrogate over 100 pair-wise interactions to establish a fitness map of these commonly observed genetic lesions in LUAD. Some genetic interactions, particularly in the context of BRAF and EGFR oncogene, have not been formally tested before with traditional GEMMs. Functional data from this work can guide the future analysis of oncogene-TSG interactions. Although the authors carried out extensive genetic screens, there was little validation or mechanism data. The authors' main conclusion is that TSGs cooperate with oncogenes in a context-specific manner. This is generally appreciated already, and the authors demonstrated this principle here in a more comprehensive manner. Thus, the main weakness of the study, in its current form, is a lack of validation data to verify the genetic interactions observed in the primary screen, and a lack of mechanistic insight on these genetic interactions. It is somewhat unclear what new knowledge has this genetic screen taught us about LUAD, and how this might deepen our understanding of the risk and treatment of human lung cancer.

We thank the Reviewer for acknowledging the rigor and comprehensiveness of our work. This Reviewer raised concerns related to the characterizing and normalizing of CRISPR-mediated gene inactivation as well as the interpretation and mechanisms of gene-gene interactions, which we have addressed through additional analyses and changes to the text. We detail these important changes to our manuscript point-by-point below.

Major points

Q2.1) The authors constructed 2 small libraries of sgRNAs targeting 28 TSGs with 1 guide per gene. It is somewhat surprising that the authors didn't attempt to validate the KO efficiencies of these sgRNAs in vitro, for example, in mouse fibroblasts or lung cancer cell lines at low MOI of < 1. Knowing all sgRNAs are effective at target knockdown will greatly inform the interpretation of neutral genetic interactions found in the tumor study. As it stands, it is unclear that when a TSG had no impact on tumor growth, whether this is simply because the sgRNA failed to KO the target gene efficiently in cells. Without sgRNA KO validation, the negative data in this study is difficult to interpret, and the authors cannot be confident at concluding many of the TSGs had no appreciable impact on tumor growth.

A2.1) The Reviewer is entirely correct that a targeted gene having no impact on tumor growth could be biological (inactivation of the gene does not have a functional consequence on tumorigenesis) or technical (inactivation of the gene was not efficient enough to see an effect). First, it is important to note that inactivation of 20 out of 28

genes had a significant and large effect on tumor size and/or relative tumor number in at least one oncogenic background. Thus, we do not need to worry about false-negative effects for these 20 genes (only vectors with sgRNAs targeting *Mga*, *Smad4*, *Ptprd*, *Atm*, *Msh2*, *Arid2*, *Kdm6a*, and *Atrx* did not significantly increase tumor size in any oncogene background). The Reviewer raises a sensible point that validating the sgRNAs *in vitro* could provide confidence that the sgRNAs effectively inactivate their intended targets. In previous publications, we generated data from *in vitro* experiments to characterize and validate sgRNAs cutting efficiency (Rogers *et al.* Nat Methods 2017¹, Supplementary Figure 6; Cai *et al.*, Cancer Discovery, 2021², Supplementary Fig. S15G–S15I). This includes 24 of the 28 sgRNAs that were used in the present study, including those targeting *Mga*, *Smad4*, *Ptprd*, *Atm*, *Arid2*, and *Atrx* (6 of the 8 that did not elicit any effects in our present study).

However, sgRNA efficiencies as assessed *in vitro* are actually poorly predictive of their effects *in vivo*. We previously quantified the cutting efficiencies of >100 sgRNAs across 50 known and putative tumor suppressor genes and found that cutting efficiency (even for different sgRNAs targeting the same gene) is poorly predictive of effects *in vivo*². This is likely due to differences in chromatin state between the *in vitro* cell lines and the lung epithelial cells from which tumors arise. Instead, we have found more value in using large amounts of *in vivo* data across many sgRNAs and targets to estimate the rate of false negatives in our data.

While we performed most of our studies with a library containing single sgRNAs targeting each of the 28 TSGs, in Supplementary Figure 5D we employed a second library, which contained two sgRNAs targeting each TSGs (the sgRNAs from the first library and a second sgRNA targeting each TSG). We found that the effects of the two sgRNAs on tumor growth were strongly correlated. Given the consistency of sgRNAs targeting genes with large effects on tumor growth, the chance that both sgRNAs targeting another gene both fail to inactivate their target and thus fail to produce a detectable biological growth effect is low. In our previous work, we performed a rigorous analysis of the expected false negative rate given that sgRNAs targeting the same gene were concordant across multiple metrics, consistent with on-target effects. We found that when one sgRNA generated a significant tumor-suppressive effect (nominal $P < 0.05$), the probability to detect a significant effect using the other guide was above 89% for all metrics assessed (Cai *et al.*, Cancer Discovery, 2021², Supplementary Table S3). Thus, the probability that both sgRNAs fail to uncover a functional tumor suppressor that has a similar effect to the TSGs identified in our analysis is below 5%. We added a paragraph to the methods (lines 487-492) describing this prior work and its connection to the data in this manuscript,

Together, these results from past *in vitro* and *in vivo* studies and the data in the present study suggest it is unlikely that functional tumor suppressors were missed for technical reasons. Nonetheless, we have focused the manuscript Results and Discussion on the positive data and have been careful not to make claims about biological significance of

neutral effects from gene KOs. We have carefully reviewed the text once again with the Reviewer's concerns on this topic in mind to confirm this point to ensure this is clear to readers of the manuscript.

Q2.2a) The analysis presented in Figure 5 and Supp Figures 9 & 10 is confusing and doesn't make great sense to me. If I understood correctly, the authors tried to "normalize" the impact of a TSG sgRNA on tumor growth by looking at its effect in Cas9NEG mice, and then deriving a ratio of its relative effect between Cas9+ and Cas9NEG mice. One would expect that none of the sgRNAs to have any effect in Cas9NEG mice. So, any impact on tumor growth in the Cas9NEG mice must be attributable to non-specific effect of a sgRNA through an CRISPR-independent mechanism. It was further confusing that the authors also switched to a different criterium for analyzing tumor fitness. In earlier analyses, the authors used a 50%, 75%, 90% and 95% tumor size to show the effect of a TSG sgRNA relative to control sgRNAs. Here the authors decided to use a relatively arbitrary tumor size cut-off to evaluate the effect of a TSG sgRNA. Why not use the same analysis as before?

A2.2a) We apologize for this confusion. This comparison does not try to "normalize the impact of a TSG sgRNA on tumor growth by looking at its effect in Cas9NEG mice" but rather corrects for difference in the titer of each TSG-targeting sgRNA relative to the sgInerts by normalizing to the tumor number data in Cas9-negative mice. Despite balancing our viral pool using *in vitro* titring, the number of tumors with each Lenti-sgRNA/Cre vector in Cas9-negative mice reflects the exact representation of each vector in the pool. Thus, this normalization is critical. This metric is described in detail in the Methods as well in Supplementary Figure 9A.

The Reviewer is also interested in why assessing both tumor size and tumor number is of value. The reason that tumor number is critical is that gene inactivation may affect tumor number independently of tumor size. For example, if inactivation of a TSG increases the rate of tumor initiation by 4-fold but does not alter the distribution of tumors sizes, this sgRNA would appear to be neutral in the tumor size percentile ratios used in Figure 2. In this manuscript and our past studies, we have found that TSGs can have different effects on tumor number and tumor size, and thus there is value in determining the effect of each gene on each aspect of tumorigenesis.

The cutoff for tumor number was chosen to be as low as technically feasible. We need to define a minimum size cutoff (number of neoplastic cells) as the number of very small tumors in our data can be driven by sequencing depth, PCR stochasticity, and Illumina error rates.

This explanation is in the first two paragraphs of the manuscript following "Genetic interactions between oncogenes and tumor suppressors impact the earliest stages of tumor development", and we have made multiple changes to the text to improve the clarity of the explanation.

Q2.2b) The data shown in Supp Figure 9A illustrates this problem. In the Cas9NEG mice, sgPten appears to strongly reduce tumor growth (0.3), whereas in the Cas9+ mice sgPten had almost no effect on tumor growth (1.1). This is in stark contrast to the data shown in Supp Figure 2C. It is unclear whether the sgPten data in these two figure panels represent the same experiment or independent repeats, or it results from a change in the data analysis method. Biologically, it might have made more sense to test the oncogene-independent effect of TSG sgRNAs in Cas9+ mice without Kras etc. Although this may result in tumor incidence too low to be informative in a library setting. Thus, it is not clear to me if the analysis in Figure 5 makes biological and technical sense, and I suggest this be removed from the paper.

A2.2b) Again, we apologize for this confusion. In Supplementary Figure 9A, the ratio of sgPten:sgInert being 0.3 is a result of the virus pool composition rather than an effect of sgPten on tumor growth (sgPten has no effect in Cas9-negative mice, which can be seen directly in Supplementary Figure 4c). We have now made this clearer in the figure legend.

Q2.3) A major weakness of the study in its current form is the lack of validation experiments beyond the in vivo library screen. The authors showed that the G12D and G12C alleles of Kras have different tumorigenic potential in this model, and the TSGs Kmt2d or Cmtr2 cooperate with G12D but not G12C to further enhance tumor growth (Figure 2). This finding, however, was not followed up by validation experiments, for example, using individual sgRNAs to examine lung tumor size and tumor proliferation/apoptosis markers. Mechanistically, how do these two genes cooperate with G12D but not G12C? In addition, in human LUAD data, do mutations these two TSGs co-occur with KRAS G12D but not KRAS G12C? Similarly, although the authors have uncovered new functional interactions with BRAF and EGFR, for example, RNF43, there are no follow-up validation experiments to further support these findings. In human LUAD, RNF43 mutation tends to co-occur with BRAF mutation, yet it tends to be mutually exclusive with EGFR mutation. Thus, validation experiments using single sgRNA would be very helpful to understand the function of RNF43 and other novel TSG-oncogene interactions.

A2.3) The Reviewer points out several important conclusions of the present study: 1) the striking difference in tumorigenic potential between Kras G12D and G12C, 2) the cooperation between Kmt2d/Cmtr2 with Kras G12D—but not with G12C—to promote tumor growth, and 3) the tumor suppressive effect of Rnf43 in both Braf- and Egfr-driven lung tumors. As the Reviewer is aware, our goal was to employ highly quantitative barcode sequencing to quantify the landscape of tumor suppression across different oncogenic contexts. We quantified tumor initiation and growth of >100 combinatorial genotypes of lung tumors, more than half of which have not previously been investigated *in vivo*. The quantitative nature of these analyses and extent of the data allowed us to make several novel discoveries regarding the relationship of the oncogenic context to tumor suppressor gene function. Thus, while individual biological effects from our study are certainly of interest, revealing their underpinnings will be best served by rigorous future mechanistic studies.

Regarding single genotype validation experiments, we agree that *in vivo* single-genotype experiments can be used effectively to uncover mechanistic insights on the effects of gene inactivation on *in vivo* tumorigenesis. We have leveraged this approach to ask a variety of questions in numerous past studies^{1,3,4}. Importantly, the overall tumor suppressive effects of several of the genes that we have uncovered with these types of multiplexed approaches have been validated by ourselves and others using conventional Cre/loxP based approaches. However, it is important to note that the tumor growth effects of a specific gene as measured within multiplexed studies (i.e. where tumors within an individual animal harbor diverse sgRNAs/genotypes) are highly consistent with the measured effects of the same gene within an equivalent single-genotype study (i.e. where all tumors within an individual animal contain the same sgRNA targeting the same single unique gene). Thus, in the context of accurately measuring the effects of gene inactivation on tumor growth *in vivo*, single-genotype experiments provide little additional confidence, are lower throughput, and require more mice to generate worse data. In the present study, we conducted many replicate multiplexed experiments across hundreds of mice to validate the effects of sgRNAs and gain confidence in the functional biological consequences of inactivating specific known and putative tumor suppressor genes (Supplementary Figure 5).

Changes in tumor growth described in this manuscript are direct quantitative measurements of the net effects of cell proliferation and death during tumorigenesis. While it is common to show BrdU and/or Ki67, these are relatively crude metrics, with the changes in proliferation rate required to increase tumor size several-fold over 12 weeks likely being impossible to detect. Our experience suggests that the analysis of cell death markers is even worse, as dead/dying cells are cleared away quickly. Follow-up experiments could involve CRISPR screens of downstream targets of genes such as *Kmt2d* or *Cmtr2* in the context of KRAS G12D and G12C tumors to understand the genetic determinants of the differential tumor suppressor activity of these genes. As noted by Reviewer #1, the present study should greatly help future studies to dissect the mechanisms of these effects. We have added a paragraph (the second to last of the Discussion) that outlines these points.

Finally, the Reviewer suggests exploring co-occurrence patterns of mutations in key tumor suppressor genes and KRAS G12D or KRAS G12C alleles in clinical LUAD data. Corroborating these associations in human data would offer additional insight into the relevance of our findings to the clinical setting. However, as we described in the manuscript, given the high tumor mutational burdens of LUAD patients with KRAS-driven tumors (which arise mostly in heavy smokers), the co-occurrence patterns of mutations in tumors with these oncogenic drivers are overwhelmed by passenger mutations, obscuring patterns driven by selection of mutations in tumor suppressor genes. This represents a statistical problem for the field and underscores the key value that direct cause and effect studies such as this bring to our understanding of the fitness landscapes of tumor suppression.

Q2.4) In this study, PTEN was identified as the strongest cooperating TSCs. The authors suggested that, together with prior studies, this result supports a major role of the PI3K pathway in human LUAD. I don't feel this argument is sound. None of the PI3K pathway genes, including PTEN and PIK3CA, are frequently mutated in human LUAD, neither have PI3K inhibitors demonstrated efficacy in LUAD in clinical studies. It is more likely that the strong synergistic effect between sgPten and Kras is an "artifact" of mouse models and reflects the fundamental difference between LUAD development in GEMM and in human. We should acknowledge such limitation of mouse models, rather than forcing the issue and use mouse models to implicate genetic interactions that are not necessarily significant in human tumors.

A2.4) Although genomic alterations in *PTEN* and *PIK3CA* are rare in human LUAD (~2.5% and 4-7%, respectively), mutations in genes that converge on the PI3K pathway occur in lung adenocarcinomas (*PTEN*, *PIK3CA*, *PIK3CG*, *PIK3R1*, etc). Importantly, there is convincing evidence that negative regulators of the PI3K pathway (*e.g.* *PTEN*) are frequently down-regulated by non-mutational mechanisms. Non-mutational inactivation may include epigenetic silencing, transcriptional repression, alternative splicing, post-translational modification, altered sub-cellular localization, and proteasome-mediated degradation^{5, 6}. For example, *PTEN* is downregulated through promoter methylation in up to a quarter of early-stage lung adenocarcinomas⁷. Many studies have reported the loss of *PTEN* protein in greater than 40% of NSCLC⁶. Thus, the frequency of genomic alterations in PI3K pathway genes in LUAD should serve as a floor, rather than as a ceiling, in estimating the true frequency of PI3K-activated tumors.

Finally, loss or decreased expression of *PTEN* is associated with poor prognosis for patients with lung cancer^{6, 8, 9} and inactivation of *PTEN* in human cancer cell lines leads to increased tumor growth (Cai *et al.*, Supplemental Figure 17h²), consistent with the strong effects of *Pten* inactivation on tumor size and number observed in our *in vivo* models.

We have revised the text to make this point clearer including the addition of new citations to provide further background and context on this specific point. We've included the relevant passage and citations here for convenience:

“Indeed, the PI3K pathway is commonly activated by non-mutational mechanism in human lung tumors, and thus *PTEN* and other members of the PI3K/AKT pathway may be important regulators of lung tumorigenesis^{5, 6, 7, 10}”

Minor points

Q2.5) The authors had constructed 2 libraries with distinct sgRNAs for each TSG. It was unclear which library was used in which experiment. Since neither library appeared to be validated for sgRNA KO efficiency, it might be hard to directly compare experimental data obtained between these two libraries.

A2.5) We used two libraries with overlapping sgRNAs targeting each gene. The library shown in Figure 1a and labeled as Lenti-D2G^{28-pool}/Cre was used in all the figures (except Supplemental Figure 5). Lenti-D2G^{28-pool}/Cre has one Lenti-sgRNA/Cre vector targeting each TSG. In the second library that was used in Supplementary Figure 5d-e, we included an additional sgRNA per target. We have now clarified the text in the Results (line 175-178) and Methods sections to make it clear that the sgRNAs are not distinct and are, therefore, comparable.

Q2.6) Does Cas9 alter tumor burden? For example, if the tumor size for inert sgRNAs are compared between Kras G12D and Kras G12D Cas9 mice (Figure 9A), is there a significant difference?

A2.6) To address this directly, we quantified the sizes of tumors in G12D and G12D/Cas9 mice. We calculated the relative size of sgInert tumors (separately for non-targeting (NT) and active-cutting (AC) sgRNAs) between G12D;Cas9 (Cas9-positive) and G12D (Cas9-negative) mice. We did this for 4 different experiments in which we had cohorts of both G12D;Cas9 and G12D Cas9-negative mice. As anticipated, Cas9 has essentially no effect on tumor size (new Supplementary Figure 4D; text in the Results at lines 301-2 and in the Methods). In Group 4, the sgInert tumors from Cas9-positive mice are slightly larger than the Cas9-negative group, but the magnitude of this difference is small, and at a level which would be considered neutral for a TSG gene (see Figure 7). This confirms that there is little to no effect of Cas9 on tumor size, which is consistent with much data from other systems suggesting that Cas9 protein in the absence of a sgRNA has no effect.

Q2.7) Figure 6B, it was unclear what the co-mutation rate (y-axis) measures. The authors seem to suggest that the stronger the genetic interaction, the more likely the TSG is to co-occur with EGFR mutation. If so, it would make more sense to plot the log2 odds ratio and q-value for co-occurrence.

A2.7) We agree that this labeling was confusing and apologize for not having made this figure panel clear in the initial submission. Co-mutation rate in Fig 6B (the y-axis) is the fraction of EGFR-mutant tumors harboring a mutation in each of the tumor suppressors. We have now relabeled the axis to “Mutation frequency in EGFR-mutant LUAD patients” to clarify this.

We are indeed suggesting that stronger genetic interactions (as measured in the GEMM system) are more likely to be observed as mutated in EGFR-mutant patients, and the figure shows this to be the case. Our understanding of the suggestion to plot log2 odds ratio and q-values would be to analyze co-occurrence of EGFR and tumor suppressors (relative to TS prevalence without EGFR). Our aim was to make comparisons with the GEMM system however, and to compare the model predictions with co-occurrence patterns in clinical data we would need a null expectation about TS prevalence in EGFR-negative patients. This is not feasible due to the confounding effects of high TMB that affects TS mutation frequencies in many other oncogenic contexts.

Q2.8) Line 351, “(Fig. 6; Supplementary Fig. 11A and B)” reference is a typo?

A2.8) We thank the Reviewer for catching this error and apologize for missing it. We adjusted the text to reference 6B instead of all of Figure 6. The reference is to the frequency of PTEN in lung adenocarcinoma patients, which is shown in Figure 6B (EGFR lung tumors), and Supp Fig. 11A and B (KRAS and BRAF, respectively).

Reviewer #3

Blair and colleagues assessed the lung tumorigenic effects of CRISPR/Cas9-mediated inactivation of 28 putative tumor suppressor genes in the context of 4 oncogenic contexts: KRAS G12D, KRAS G12C, BRAF V600E, and EGFR L858R. To date, this study likely represents the largest, most comprehensive in vivo analysis of cross-oncogene tumor suppressor effect comparisons. The authors conclude that the fitness landscape is rugged (i.e. the effect of tumor suppressor inactivation often switches between beneficial and deleterious depending on the oncogenic context) and shows no evidence of diminishing-returns epistasis within variants of the same oncogene. The findings presented here suggest that a simple linear oncogenic signaling relationship of EGFR to KRAS to BRAF does not exist because off-axis signaling likely contributes to determining the fitness effects of inactivating tumor suppressors.

This manuscript is well-written and represents a complete narrative that has been strengthened by comprehensive, well-controlled experiments. The data is presented nicely, and the scientific rigor is robust.

We thank the Reviewer for the positive feedback and for recognizing the depth, comprehensiveness, and robustness of our work. We are pleased that the Reviewer found our narrative complete and the data well-presented. The acknowledgment of the potential importance of our findings in elucidating the complexity of oncogenic signaling relationships in cancers is highly encouraging. We have taken careful note of the Reviewer's comments below and addressed each through additions to the Supplementary Figures and manuscript text.

Q3.1) In Figure 5C, what was the rationale for analyzing Braf;Cas9 relative to Cas9negative (G12D) as opposed to Braf;Cas9 relative to Cas9negative (Braf) as done in panels 5A, 5B, and 5D for G12C, G12D, and Egfr, respectively?

A3.1) Cas9negative mice were used only to quantify the proportions of each virus in a virus pool, and, for this purpose, any oncogenic background of Cas9negative mice functions equivalently. Each of the four cohorts presented in Figure 5 was transduced with a distinct virus pool, and so each required their own Cas9negative cohort. For G12D;Cas9, G12C;Cas9, and EGFR;Cas9, we happened to use a Cas9negative cohort with a matched oncogene, while for BRAF;Cas9 we used Cas9negative (G12D). We have made several adjustments to the paragraph in which Fig. 5A-D are introduced to make this clearer.

Q3.2) Methods: Please include much greater details regarding the intratracheal delivery of the lentiviral to the mice, including anesthesia methods, lentivirus dose, numbers of mice, age of mice, etc. The current description is not comprehensive enough that other researchers could replicate it based on the methodology outlined.

A3.2) We apologize for not making this aspect of the methodology clearer in the initial submission. We have revised the Methods to include the relevant citations to enable other researchers in the field but unfamiliar with the intratracheal lentiviral delivery method to be able to learn and replicate it. Importantly, we have included a citation to an article that describes in detail the intratracheal intubation method for the application of delivering lentiviral-Cre vectors to conditional mouse lung cancer models¹¹. This citation should have been included in the original submission and we regret the oversight. Moreover, we added a new supplemental table to the manuscript (see Supplementary Data 1) that outlines relevant metadata, such as genotype, sex, age, and administered lentiviral titer, for the mice used in this specific study. We have added to the Methods section that the mice were anesthetized with isoflurane.

Q3.3) The Discussion section currently lacks depth and requires greater assessment of the findings presented here within the context of existing knowledge. How can the conclusions of this manuscript translate to improved therapies for lung cancer patients? What opportunities for future investigation do these results create? What are the limitations of this current study and how could they be overcome?

A3.3) We thank the Reviewer for suggesting these ways of strengthening the Discussion section of our manuscript. We have now added additional text towards the end of this section that includes elaborations on each suggestion. To address the implications for therapeutics, we discuss how these results suggest a possible explanation for low response rates to many therapies and how this platform can and has been used to reveal genetic sources of drug resistance, ultimately leading to improved outcomes. To address opportunities and limitations, we discuss ways for future studies to extend our work to better reveal mechanisms underlying the observed fitness effects.

The study by Blair and colleagues used various autochthonous mouse models of lung adenocarcinoma (LUAD) driven by KrasG12D, KrasG12C, BrafV600E and EgfrL858R, barcoded lenti-sgRNA/Cre vectors targeting in each of the four models 28 known and putative tumor suppressors, and the tuba-seq previously developed by the group and which quantifies independent tumor clones by deep sequencing of barcodes present in each genomically-integrated lentivirus. The study found that KrasG12D leads to larger and faster lung tumor development, irrespective of the inactivation of tumor suppressors. Inactivation of various tumor suppressors had, for the most part, similar effects on KrasG12D and KrasG12C tumors -- although they did find some disparate functional effects for some, Kmt2d loss enhanced KrasG12D pathogenesis. Braf and Egfr mutant tumors had distinct effects on tumor development - Braf mutant mice had higher burden than those with mutant Egfr and their size were more or less uniform. The authors conclude that the four drivers have distinct tumor developmental potential with KrasG12D displaying the highest potential. Interestingly, loss of some tumor suppressors (e.g. Lkb1) showed opposing effects between Kras and Egfr mutant tumors - Lkb1 loss increased tumor size in KrasG12D models but decreased tumor size in Egfr mutant mice. Similar analysis was performed when comparing to Cas9 negative cells to determine effects of tumor suppressor loss on early stages of tumor development. Interestingly, t least in Egfr mutant tumors, tumor sizes by tumor suppressor loss correlated with co-mutation rate of the tumor suppressor (e.g., Lkb1, Keap1) with EGFR in human LUAD patients. On the other hand, Pten loss yields similar effects on tumor development across different oncogene-driven models. Lastly, the study summarizes effect of tumor suppressor loss on tumor size and number in the context of activation of the four oncogenic drivers.

The study will pique interest in the field of LUAD development since it highlights oncogene-tumor suppressor interactions that would not be readily observed/interrogated in human LUAD. For instance, Pten is very rarely co-mutated with Kras in human tumors, and the study clearly demonstrates tumor promoting effects of Pten loss in mouse models with mutant Kras activation. There are some comments that need to be addressed prior to publication for the astute readership of Nature communications.

We thank the Reviewer for their thorough and insightful assessment of our study. We are encouraged by the recognition that our work will be of interest to the field and highlight important oncogene-tumor suppressor interactions that might not be readily detectable from human data alone. We appreciate the Reviewer's comments and suggestions below on ways to increase the value of our manuscripts to the discerning Nature Communications readership. We have carefully reviewed these comments and addressed each of them through additional analyses, experiments, and revisions to the text. We elaborate on these revisions and the ways in which we have strengthened the manuscript in response to the Reviewer's comments point-by-point below.

Q4.1) It is not clear why the study did not first use animals with control lenti-sgRNA to compare and contrast tumor development between KrasG12D and KrasG12C mice.

A4.1) We agree with the Reviewer that to compare and contrast the oncogenic potential of Kras G12D- and Kras G12C-driven tumors, we could have initially conducted experiments in which tumors were initiated in *G12D* and *G12C* mice with a barcoded Lentiviral-Cre vectors. In this context, tumors within a single mouse would only have G12D- or G12C-driven tumors. Although this would have been the simplest approach to address this specific question, we have performed many experiments in these models over the past decade and have consistently found that the tumor growth effects of a specific genotype as measured within multiplexed studies (i.e. where tumors within an individual animal harbor diverse sgRNAs/genotypes) are highly consistent with the measured effects of the same gene within an equivalent single-genotype study (i.e. where all tumors within an individual animal contain the same sgRNA targeting the same single unique gene). We also have deep experience extracting the relevant data on a single tumor genotype (i.e. KRAS G12D or G12C tumors with no additional alterations) from a multiplexed data pool. Thus, because we were interested in comparing both the baseline oncogenicity of G12D and G12C as well as overlaying the map of tumor suppressive effects on top of these oncogenes, we decided to directly begin with multiplexed experiments. This had the advantage of producing internally controlled data for the driver oncogenes and also tumor suppressor genes, and greatly reduced the number of mice needed for these studies.

Q4.2) On that theme, there appears to be no difference in tumor burden when assessing KrasG12C/D mice with control lenti-sgRNA or sgRNA targeting the 28 tumor suppressors - perhaps suggesting that few tumor suppressors significantly impact tumor initiation by oncogenic Kras. Indeed this appears to be the case upon inspection of individual tumor suppressor sgRNAs (e.g., Figure 2). The authors need to discuss this point.

A4.2) The total tumor burden (normalized to viral titer) in Figure 1 is in fact greater than the sgInert tumor burden, although the difference is difficult to see in comparison to the very large difference between G12D;Cas9 and G12C;Cas9 mice. In G12D;Cas9 mice, at the 15 week timepoint, the ratio of total tumor burden cells (normalized to viral titer) to sgInert tumor burden is 1.85. In G12C;Cas9 mice at the 15 week timepoint, the ratio of total tumor burden (normalized to viral titer) to sgInert tumor burden is 1.37. To clarify this to readers, we have added text at lines 136-140.

The total tumor burden reflects an average effect of tumor suppressor strength that is difficult to interpret given that it is dependent on the composition of the virus pool. When the effect on burden of each individual vector is calculated, the effects on burden are similar to the effects on growth (see Reviewer Figure 1 below), as seen in Figure 2 and Figure 7. As the reviewer points out, there are several genes that do not have a strong effect on tumor burden or size, although there are also genes that when inactivated increase the tumor burden by up to 32-fold.

Of additional note, the strongest growth effects occur in the tails of the distribution, which can be seen in the relative tumor size plots in Figure 2. The measured growth

effect is larger at higher percentiles of the tumor size distribution, which indicates that the largest tumors in each genotype are growing the most when compared to sgInert tumors. These large tumors contribute to increased tumor burden, but we do not expect the increase in tumor burden when summed over all tumor sizes to be as large in magnitude as the growth effect measured for those large tumors alone.

Reviewer Figure 1. Effect of tumor suppressor inactivation on tumor burden.

a-d. Impact of inactivating each gene on tumor burden in the indicated genotypes of mice. Relative burden is calculated by dividing the ratio of neoplastic cells of Lenti-sgRNA/Cre tumors to neoplastic cells of Lenti-sgInert/Cre tumors in Cas9-positive mice by the same ratio in mice without Cas9 that had been initiated with the same pool of viruses.

Q4.3) The authors conclude that *Braf* mutation, relative to mutant *Egfr*, leads to more tumor numbers. There is inconsistency between panels C and say E in figure 3. The histopathological images in panel C suggest high tumor number and burden in *Egfr*;Cas9 mice.

A4.3) We apologize that Figure 3C in our initial submission led to some confusion and we thank the Reviewer for bringing this to our attention. As the Reviewer alluded to, the top row of panel 3C contains H&E images of tumor-bearing lung lobes from *Braf*;Cas9 and *Egfr*;Cas9 mice. Although the depicted *Braf*;Cas9 lung section in this panel contains ~2x more visible tumors than the *Egfr*;Cas9 lung section, a likely source of confusion is that the virus titer delivered to each genotype of mouse represented in this panel was drastically different. The depicted *Braf*;Cas9 mice received 900,000 TUs, while the

Egfr;Cas9 mice received 5,000,000 (>5x more virus). Precisely because of the difference in the tumorigenic capacity of oncogenic *Egfr* and *Braf*, each genotype of mice was transduced with a different titer to achieve optimal tumor number and burden for our studies (i.e. maximizing tumor formation without generating mouse morbidity by the study end point). Our intention in including these histological images was to provide readers with a sense of the qualitative differences in the *Braf*- and *Egfr*-driven tumors, for instance that the sizes of *Braf*-driven are very consistent, while the sizes of *Egfr*-driven tumors are highly variable. Another important point is that only the largest tumors are visible by eye within the histological images, so observations derived from this panel are only expected to be consistent with the far-right tail of tumor size distributions derived from tumor barcode sequencing. The viral titer differences between the *Braf;Cas9* or *Egfr;Cas9* mice were initially only included in the figure legend. We have now included those titers directly within the Figure 3C, so this will be clear to readers.

Q4.4) While the use of the tuba-seq platform is commendable and useful, there are lost opportunities to determine how loss of tumor suppressors not only impacts tumor number and size but also the histological grade/stage of the lesion (hyperplasia versus adenoma versus adenocarcinoma). It will be important to determine by comprehensive histopathological analysis how tumor suppressor loss impacts pathologic progression of lung lesions in the context of the four oncogenic drivers.

On that theme it is important to validate in resultant (if available and not entirely sequenced, say in additional mice) key markers (e.g., phosphorylated ERK in case of *Kras* signaling) that may inform of the mechanisms underlying effects of tumor suppressor loss on oncogene-driven lung tumor development.

A4.4) We thank the Reviewer for these valuable points. As the Reviewer is aware, our goal was to employ highly quantitative barcode sequencing to quantify the landscape of tumor suppression across different oncogenic contexts. We quantified tumor initiation and growth of >100 combinatorial genotypes of lung tumors, more than half of which have not previously been investigated *in vivo*. This led to several unexpected results that we believe are both fundamentally important and novel.

Our approach generated many genotypes in parallel and uses the entire tumor-bearing lungs for DNA extraction and barcode sequencing, and we therefore do not have material left for histological analysis (and even if we did, we would not know the genotypes of individual tumors). In the future, being able to integrate our multiplexed approach with histological grading, pERK staining, and other more mechanism-focused data like spatial transcriptomics would be of high value. These are critical points to emphasize, and we have added a paragraph in the Discussion on the value of broad multiplexed screens, how they are complementary to single-genotype studies that facilitate explorations of mechanism, and suggests the later as a follow-up to this work. Nevertheless, we hope that the Reviewer will appreciate that our work stands on its own to reveal the broad contours of the fitness landscape.

References

1. Rogers ZN, *et al.* A quantitative and multiplexed approach to uncover the fitness landscape of tumor suppression in vivo. *Nature methods* **14**, 737-742 (2017).
2. Cai H, *et al.* A functional taxonomy of tumor suppression in oncogenic KRAS-driven lung cancer. *Cancer discovery* **11**, 1754-1773 (2021).
3. Rogers ZN, *et al.* Mapping the in vivo fitness landscape of lung adenocarcinoma tumor suppression in mice. *Nature Genetics* **50**, 483-486 (2018).
4. Foggetti G, *et al.* Genetic determinants of EGFR-driven lung cancer growth and therapeutic response in vivo. *Cancer discovery* **11**, 1736-1753 (2021).
5. Dillon LM, Miller TW. Therapeutic targeting of cancers with loss of PTEN function. *Curr Drug Targets* **15**, 65-79 (2014).
6. Gkoutakos A, *et al.* PTEN in Lung Cancer: Dealing with the Problem, Building on New Knowledge and Turning the Game Around. *Cancers (Basel)* **11**, (2019).
7. Soria JC, *et al.* Lack of PTEN expression in non-small cell lung cancer could be related to promoter methylation. *Clin Cancer Res* **8**, 1178-1184 (2002).
8. Xiao J, *et al.* PTEN expression is a prognostic marker for patients with non-small cell lung cancer: a systematic review and meta-analysis of the literature. *Oncotarget* **7**, 57832-57840 (2016).
9. Fischer T, *et al.* PTEN mutant non-small cell lung cancer require ATM to suppress pro-apoptotic signalling and evade radiotherapy. *Cell Biosci* **12**, 50 (2022).
10. Ocana A, *et al.* Activation of the PI3K/mTOR/AKT pathway and survival in solid tumors: systematic review and meta-analysis. *PLoS One* **9**, e95219 (2014).
11. DuPage M, Dooley AL, Jacks T. Conditional mouse lung cancer models using adenoviral or lentiviral delivery of Cre recombinase. *Nature protocols* **4**, 1064-1072 (2009).

REVIEWERS' COMMENTS

Reviewer #2 (Remarks to the Author):

In the revision, the authors provided clarification and additional details on methods for data analysis. The authors did not present much in terms of new data and further experiments to address the questions I raised. The revision is therefore largely similar to the original manuscript, and does not represent a substantial improvement.

Reviewer #3 (Remarks to the Author):

The authors have addressed the reviewers' critiques well. In my opinion, this manuscript is acceptable for publication and no additional revisions are needed.

Reviewer #4 (Remarks to the Author):

The authors did a good job addressing the previous critiques. The manuscript is improved.